# Understanding the Differences in Foundation Models: Attention, State Space Models, and Recurrent Neural Networks

**Jerome Sieber**[*]
ETH Zurich
Zurich, Switzerland
jsieber@ethz.ch

**Carmen Amo Alonso**[*]
ETH Zurich
Zurich, Switzerland
camoalonso@ethz.ch

**Alexandre Didier**
ETH Zurich
Zurich, Switzerland
adidier@ethz.ch

**Melanie N. Zeilinger**
ETH Zurich
Zurich, Switzerland
mzeilinger@ethz.ch

**Antonio Orvieto**
ELLIS Institute Tübingen
Tübingen, Germany
antonio@tue.ellis.eu

## Abstract

Softmax attention is the principle backbone of foundation models for various artificial intelligence applications, yet its quadratic complexity in sequence length can limit its inference throughput in long-context settings. To address this challenge, alternative architectures such as linear attention, State Space Models (SSMs), and Recurrent Neural Networks (RNNs) have been considered as more efficient alternatives. While connections between these approaches exist, such models are commonly developed in isolation and there is a lack of theoretical understanding of the shared principles underpinning these architectures and their subtle differences, greatly influencing performance and scalability. In this paper, we introduce the Dynamical Systems Framework (DSF), which allows a principled investigation of all these architectures in a common representation. Our framework facilitates rigorous comparisons, providing new insights on the distinctive characteristics of each model class. For instance, we compare linear attention and selective SSMs, detailing their differences and conditions under which both are equivalent. We also provide principled comparisons between softmax attention and other model classes, discussing the theoretical conditions under which softmax attention can be approximated. Additionally, we substantiate these new insights with empirical validations and mathematical arguments. This shows the DSF's potential to guide the systematic development of future more efficient and scalable foundation models.

## 1 Introduction

Foundation models serve as the backbone for a wide range of tasks across Artificial Intelligence due to their ability to learn complex interactions in large datasets [1]. In recent years, the attention mechanism [2] has been the dominating token-mixing strategy in foundation models. However, its major computational bottleneck, i.e., the quadratic complexity with context length, has posed a challenge to scaling and deploying these models beyond moderate context lengths [3]. In order to mitigate these issues, attention-free architectures have been proposed: prominent examples of these are the novel State Space Models (SSMs) [4–8], as well as recent efforts to enhance Recurrent Neural Networks (RNNs) [9–12]. Although these models show great promise in boosting efficiency, efforts

---

[*]These authors contributed equally; ordered randomly.

38th Conference on Neural Information Processing Systems (NeurIPS 2024).

to provide a rigorous theoretical comparison are scarce, and current comparisons with attention are merely empirical (see Section 5 for an in-depth discussion). Despite the prevalence and ubiquity of foundation models, a principled understanding of the similarities and differences among these different design strategies is currently lacking.

In order to close this gap, we introduce the Dynamical Systems Framework (DSF) – a theoretical framework based on a control theoretic perspective – that allows us to evaluate the similarities and differences between different foundation models in a principled manner. The DSF serves as a powerful tool for approaching theoretical research questions about foundation models, enabling direct comparisons – both theoretical and experimental – across architectures such as attention mechanisms, SSMs, and RNNs. We believe that the DSF provides new insights on the most relevant features found in current architectures, and can inform a systematic development of future hybrid models. The DSF further simplifies identification of existing computational algorithms to apply to newly developed models. Rather than providing an exhaustive list of insights, the results included below are meant to exemplify important questions that the DSF can answer and guide future research. Specifically, we explore the following questions:

- **How are attention mechanisms, SSMs, and RNNs related?**
  *TL;DR:* All three model classes can be represented as recurrent models, which can be compared using the proposed DSF.

- **Can softmax attention be expressed as a recurrent model?**
  *TL;DR:* Softmax attention translates to a recurrent model within the DSF, however the hidden state dimension needs to be infinite.

- **Why does state expansion help to improve performance of RNNs and SSMs?**
  *TL;DR:* This is related to the second question: state expansion increases the dimension of the hidden state thus allowing for an increased expressivity of the model (Lemma 2).

- **Why do SSMs significantly outperform attention on the LRA benchmark?**
  *TL;DR:* The performance gap can be explained by the recurrent normalization strategy (discretization step) used by selective SSMs as discussed in Section 4.2.

- **How closely are linear attention, S6 (i.e. Mamba) related?**
  *TL;DR:* The common feature is the coupling of state transition and input matrix via a single (normalization) parameter in their recurrent representation. However, the models differ in the parameterization of this parameter, which we analyze experimentally.

- **What do selective SSMs teach us about improving RNN architectures?**
  *TL;DR:* Replacing the state transition in a RNN variant - qLSTM - with the state transition of S6 improves performance of the RNN.

We note that the main contribution of our paper is the introduction of the DSF, which is a unifying framework for analysis of attention mechanisms, SSMs, and RNNs. To the best of our knowledge, this is the first unified framework that allows analysis of all three model classes in the same parameterization and thus allows to identify differences in the models that lead to significant performance improvements. While some of the provided results already exist in the literature (e.g that increased state size improves performance), we also provide novel insights unique to the DSF framework in a comprehensive way that enables further analysis with control theoretical tools.

**Notation:**   We use Latin letters in the following way: $N$ is the size of the hidden state in the DSF, $n$ the state expansion, $d$ the embedding size or model size, and $L$ the sequence length. A visual representation of these dimensions is given in Appendix A. We use superscripts, e.g. $\cdot^d$, to denote the elements or block-elements of a matrix and a block-matrix. We use subscripts, e.g. $\cdot_i$, to denote the time index (or input dependency). Specifically, $v_i$ represents the value of vector $v$ at time $i$. We use bold notation to indicate sequences, i.e., $\mathbf{v}_i = [v_1, \ldots, v_i]$. We use $\sigma(\cdot)$ to denote the sigmoid function. The products $\odot$ and $\otimes$ denote the Hadamard (element-wise) product and the Kronecker (block-wise) product, respectively. $\mathbb{I}_n$ denotes the identity matrix of size $\mathbb{R}^{n \times n}$. Generally, we omit stating the bias term for weight matrices unless stating the bias term helps with clarity.

## 2 Preliminaries

In this section, we introduce the key architectural components studied in this work: attention, SSMs, and RNNs. We remark that these components are often the central block - considered to be the backbone - within a complex architecture composed of other blocks and skip connections (see for instance [13]). In what follows, we review exclusively the backbone block, which we denote as $f(\cdot)$ in $\mathbf{y} = f(\mathbf{u})$, where $\mathbf{u} \in \mathbb{R}^{L \times d}$ and $\mathbf{y} \in \mathbb{R}^{L \times d}$ are the input and output sequences, respectively.

### 2.1 Attention

The standard self-attention block [2] consists of three matrices: $W_Q$, $W_K$, and $W_V$, which are the learnt parameters of the model. These matrices, when multiplied with the input $\mathbf{u}$, yield the queries $\mathbf{q} \in \mathbb{R}^{d_k}$, keys $\mathbf{k} \in \mathbb{R}^{d_k}$, and values $\mathbf{v} \in \mathbb{R}^{d_v}$, respectively:

$$\mathbf{q} = \mathbf{u}W_Q, \quad \mathbf{k} = \mathbf{u}W_K, \quad \mathbf{v} = \mathbf{u}W_V. \tag{1}$$

Keys, queries, and values are then combined in the attention block to produce the output

$$\mathbf{y} = \zeta\left(\frac{\mathbf{q}\mathbf{k}^\top}{\sqrt{d_k}}\right)\mathbf{v}, \tag{2}$$

where $\zeta(\cdot)$ is a map $\mathbb{R}^L \to \mathbb{R}^L$ and is applied row-wise. For standard self-attention, the softmax function is used, i.e. $\zeta(\cdot) = \text{softmax}(\cdot)$, but given the limitations of the softmax function, alternative formulations have been proposed. We consider two formulations of attention: softmax attention (2) and linear attention [14]. We focus on masked attention formulations, i.e., the attention matrix $\zeta(\mathbf{q}\mathbf{k}^\top)$ has a lower-triangular structure, and to simplify the derivations, we drop the scaling factor $\sqrt{d_k}$.

### 2.2 State Space Models

Architectures based on a state space parametrization compute the output $\mathbf{y}$ through a dynamic recurrence of input signals at each time step $i$, i.e.,

$$h_i = A_i h_{i-1} + B_i u_i \tag{3a}$$
$$y_i = C_i h_i + D_i u_i, \tag{3b}$$

where $h_i$ is the hidden state of the system, and the dynamic matrices of appropriate dimensions $A_i, B_i, C_i, D_i$ are the learnt model parameters. Different time-varying and time-invariant parameterizations for $A_i, B_i, C_i, D_i$ have been proposed in the literature (an overview is given in [15]). Here we discuss the most prominent one.

**S6.** The first selective SSM parametrization (S6) was introduced together with the Mamba architecture [7]. The S6 block parametrizes the recurrence as

$$A_i = e^{-\Delta_i A}, \qquad B_i = \Delta_i W_B u_i, \qquad C_i = W_C u_i, \qquad D_i = W_D u_i, \tag{4}$$

with $\Delta_i = \text{softplus}(W_\Delta(W_u u_i) + b_\Delta)$ for every $i$, $W_\Delta$, $W_u$, $W_B$, $W_C$, $W_D$, and $A$ are learnt matrices of appropriate dimensions, and $b_\Delta$ is a learnt bias. While SSM models allow for complex-valued matrices $A_i, B_i, C_i, D_i$, here we restrict ourselves to real-valued matrices as in [7].

### 2.3 Recurrent Neural Networks

Similar to SSMs, RNNs also parameterize the input-output relationship via a recurrent computation, commonly given by the long short-term memory (LSTM) [16], i.e., at each time step $i$

$$x_i = f_i \odot x_{i-1} + i_i \odot \bar{u}_i, \tag{5a}$$
$$y_i = o_i \odot \tanh(x_i), \tag{5b}$$

where $\bar{u}_i$ represents the pre-processed raw input $u_i$, i.e.,

$$\bar{u}_i = \tanh(W_u u_i + U_u y_{i-1}), \tag{6}$$

and $f_i$, $i_i$, and $o_i$ are the forget gate, the input gate, and the output gate, respectively,

$$f_i = \sigma(W_f u_i + U_f y_{i-1}), \quad i_i = \sigma(W_i u_i + U_i y_{i-1}), \quad o_i = \sigma(W_o u_i + U_o y_{i-1}), \tag{7}$$

where $W_f, W_i, W_o$ and $U_f, U_i, U_o$ are the learnt gate parameters. In this paper, we focus on two variants: quasi LSTMs (qLSTM) [9], which removes the output dependence of the gates, and RG-LRU [10], which attempts to integrate ideas from SSMs into RNNs.

**qLSTM.** The qLSTM model is parameterized by recurrence (5) with pre-processed input $\bar{u}_i$ and gates $f_i$, $i_i$, $o_i$:

$$\bar{u}_i = \tanh(W_u u_i), \quad f_i = \sigma(W_f u_i), \quad i_i = \sigma(W_i u_i), \quad o_i = \sigma(W_o u_i). \tag{8}$$

**RG-LRU.** The RG-LRU model presents a hybrid between a qLSTM and a SSM using the recurrence

$$x_i = a_i \odot x_{i-1} + \sqrt{1 - a_i^2} \odot (i_i \odot u_i) \tag{9a}$$

$$y_i = x_i, \tag{9b}$$

with the following gates and no pre-processing of $u_i$:

$$r_i = \sigma(W_a u_i), \quad i_i = \sigma(W_u u_i), \quad a_i = e^{-cr_i \odot \mathrm{softplus}(\Lambda)}. \tag{10}$$

## 3 Dynamical Systems Framework for Architecture Comparison

In this section, we introduce the Dynamical Systems Framework (DSF) that allows in-depth analysis of the architectural features of attention, SSMs, and RNNs from a dynamical systems perspective. We use this to rewrite the parametrizations in a common framework and provide detailed comparisons.

### 3.1 Dynamical Systems Framework (DSF)

The DSF relies on a dynamical systems representation of the architectures. A dynamical system models how a system's state, here denoted by $h$, evolves over time according to a difference or differential equation. Dynamical systems often evolve under the evolution of some input $u$, and the observable is an output $y$. These systems capture time-dependent processes, rendering them suitable for understanding the behavior of sequence models. Here, we choose a recurrent state space representation. This choice is motivated by the widespread use of state space model representations for dynamical systems. Moreover, we show in later sections that this representation encompasses attention, RNNs, and SSMs in a suitable fashion that allows for further analysis. In particular, a linear structured time-varying (LTV) dynamical system is defined by the recurrence

$$h_i = \Lambda_i h_{i-1} + B_i u_i \tag{11a}$$

$$y_i = C_i h_i + D_i u_i, \tag{11b}$$

where $h_i \in \mathbb{R}^N$ is the hidden state initialized with $h_{-1} = 0$, $\Lambda_i \in \mathbb{R}^{N \times N}$ is the diagonal state transition matrix, $B_i \in \mathbb{R}^{N \times d}$ and $C_i \in \mathbb{R}^{d \times N}$ are the input and output matrices, respectively, and $D_i \in \mathbb{R}^{d \times d}$ is a scaled skip connection. Dynamical system (11) can alternatively be written in its convolutional representation, i.e., $\mathbf{y} = \Phi \mathbf{u}$, where the convolutional kernel $\Phi$ is defined as

$$\Phi = \begin{bmatrix} C_0 B_0 + D_0 & & & \\ C_1 \Lambda_1 B_0 & C_1 B_1 + D_1 & & \\ \vdots & \ddots & \ddots & \\ C_L \prod_{k=1}^{L} \Lambda_k B_0 & \dots & C_L \Lambda_L B_{L-1} & C_L B_L + D_L \end{bmatrix}. \tag{12}$$

Note that the convolution kernel $\Phi$ is of the same dimension as the attention matrix $\zeta(\mathbf{q}\mathbf{k}^\top)$ and that these matrices are equivalent, up to the scaling factor $W_V$ used in self-attention.

**Remark 1.** *This recurrent view yields a causal convolution kernel by definition. However, certain models (e.g. non-masked attention) also use non-causal kernels. This can be incorporated in the DSF (11) by modifying the state update (11a). For the sake of simplicity and consistency with the recent literature, we stick with causal models in the following.*

### 3.2 Architecture Reformulation

In the following, we show how popular architectures based on attention, SSMs, and RNNs can be rewritten into the DSF. To do this, all models will be reformulated into recurrence (11), i.e., all resulting DSF representations will have hidden state dimension $N$.[2] Although the parametrization

---

[2]The dimensions used in the following are visualized in Appendix A for clarity.

of models commonly found in the literature is conductive to efficient computation, here we depart from this convention. The goal of the DSF reformulation is to establish a theoretical framework that leads us to mathematical insights on the design of these models. The presented formulations are not intended to be computationally implemented in DSF form, however the framework can be used to identify computational algorithms for new architectures. For instance, the convolutional form of linear attention (12) is efficiently implemented via flash linear attention [17]. However, using the recurrent form derived below it can also be implemented via scan algorithms [18], e.g., parallel scan [5, 6] or accelerated scan [19]. Given that the structural requirements on the model parameterization of the algorithm is met, the DSF allows to identify existing algorithms to apply to new models even if the algorithm was designed for another model class.

### 3.2.1 Attention

In the following, we assume that we can separate the nonlinear map in (2) as

$$\zeta(q_i^\top k_j) = \frac{\phi(q_i)^\top \psi(k_j)}{\eta(q_i, \mathbf{k}_i)}, \tag{13}$$

where $\phi(\cdot) : \mathbb{R}^m \to \mathbb{R}^n$, $\psi(\cdot) : \mathbb{R}^m \to \mathbb{R}^n$, and $\eta(\cdot, \cdot) : \mathbb{R}^m \times \mathbb{R}^{m \times (i+1)} \to \mathbb{R}$, which is the case for all the considered architectures in this paper. Note that if $\zeta(\cdot)$ is a kernel function, the proposed separability is satisfied by construction, as it holds that $\phi = \psi$ and $\eta = 1$. This allows us to write the self-attention input-output relationship as

$$y_i = \sum_{j=0}^{i} \frac{\phi(q_i)^\top \psi(k_j)}{\eta(q_i, \mathbf{k}_i)} W_V u_j, \tag{14}$$

with $q_i = W_Q u_i \in \mathbb{R}^m$, $k_j = W_V u_j \in \mathbb{R}^m$, and $W_Q \in \mathbb{R}^{m \times d}$, $W_K \in \mathbb{R}^{m \times d}$, $W_V \in \mathbb{R}^{d \times d}$. Hence, equation (14) can be reformulated into the DSF (11) as a dynamical system of dimension $N = nd$, i.e., with hidden state $h_i \in \mathbb{R}^{nd}$, and dynamic matrices

$$\Lambda_i = \frac{\eta(q_{i-1}, \mathbf{k}_{i-1})}{\eta(q_i, \mathbf{k}_i)} \mathbb{I}_{nd} \in \mathbb{R}^{nd \times nd}, \tag{15a}$$

$$B_i = \left( \frac{1}{\eta(q_{i-1}, \mathbf{k}_{i-1})} \mathbb{I}_d \otimes \psi(k_j) \right) W_V \in \mathbb{R}^{nd \times d}, \tag{15b}$$

$$C_i = \mathbb{I}_d \otimes \phi(q_i)^\top \in \mathbb{R}^{d \times nd}. \tag{15c}$$

We note that for the recurrence (11), the matrix $\Lambda_i$ is given as an $nd \times nd$ matrix, where $n$ is the number of features in $\phi$ and $\psi$, and $d$ is the input dimension. However, due to the scalar structure of $\Lambda_i$ in (15a), it can be implemented as the scalar multiplication $\frac{\eta(q_{i-1}, \mathbf{k}_{i-1})}{\eta(q_i, \mathbf{k}_i)} h_{i-1}$ in (11). Hence, the hidden state is never materialized as such in the computation of the attention scores. Interested readers are referred to Appendix B for a detailed derivation.

**Linear Attention.** In the case of *linear attention*, both maps $\phi(\cdot)$ and $\psi(\cdot)$ in the DSF parametrization (15) are separable and we use the kernel proposed in [14], i.e.,

$$\phi(q_i) = \text{elu}(q_i) + 1, \quad \psi(k_j) = \text{elu}(k_j) + 1, \quad \eta(q_i, \mathbf{k}_i) = (\text{elu}(q_i) + 1) \sum_{j=0}^{i} (\text{elu}(k_j) + 1), \tag{16}$$

where $\text{elu}(\cdot)$ is the exponential linear unit.

**Generalized Linear Attention.** We also study *generalized* linear attention, where we require that the maps $\phi(\cdot)$, $\psi(\cdot)$ are linear, but allow for general nonlinear normalization functions $\eta(q_i, \mathbf{k}_i)$, i.e.,

$$\phi(q_i) = q_i, \quad \psi(k_j) = k_j, \quad \eta(q_i, \mathbf{k}_i). \tag{17}$$

**Softmax Attention.** Softmax attention also satisfies the assumption of separability (13). However, it holds that the feature vector representation of the transformed Gaussian kernel in the softmax function, i.e., $e^{q_i^\top k_j}$, is infinite dimensional. Hence, the DSF representation (15) of softmax attention (2) and its corresponding hidden state dimension $N$ would also be infinite dimensional. This insight gives further motivation to approximations of the softmax function by using, e.g., a Taylor series approximation such as in [20], to render the feature vector finite-dimensional.

**Lemma 1.** *Softmax attention* (2) *can be expressed by separable attention* (13) *with*

$$\phi(q_i)^\top \psi(k_j) = \phi(q_i)^\top \phi(k_j) = e^{q_i^\top k_j}, \quad \eta(q_i, \mathbf{k}_i) = \sum_{j=0}^{i} e^{q_i^\top k_j}, \tag{18}$$

*where* $\phi(q_i) := c \cdot \left[1, q_i, \bigotimes_{j=1}^{2} q_i, \bigotimes_{j=1}^{3} q_i, \ldots\right]$ *is an infinite-dimensional feature vector and c is a matrix of constant coefficients.*

*Proof.* The exponential in softmax attention $e^{q_i^\top k_j}$ can be expressed in terms of its Taylor expansion, which consists of an infinite sum of polynomial kernels of increasing degree $p$, decomposable through the vectors of monomials $\bigotimes_{j=1}^{p} q_i$. See Appendix C for a complete proof. □

The work in [21] analyzes softmax attention as a kernel smoother and [14] shows that a kernel-based formulation can lead to linear complexity in sequence length for finite dimensional kernels. In [22], a kernel-based formulation of softmax is used to propose orthogonal random features to model softmax attention with linear complexity. In [20] a Taylor approximation of softmax attention is proposed, also leading to linear complexity. Finally, [23] relates transformer decoders to dynamical systems with increasing state size arising from the masked upper triangular part of the attention matrix. Compared to these works, we analyze how the proposed formulations compare in the recurrence (11) allowing us to compare to SSMs and RNNs in the following sections.

### 3.2.2 State Space Models

SSM models are straightforward to rewrite in the DSF given their intrinsic recurrent linear representation. However, similarly to attention, we slightly rewrite the standard representation introduced in the literature to reveal deeper insights obscured by the standard representation focused on computational efficiency. The detailed derivation can be found in Appendix E.

**S6.** The S6 parametrization can be written in the DSF (11) as

$$\Lambda_i = e^{-(\Delta_i \otimes \mathbb{I}_n) \odot A} \in \mathbb{R}^{nd \times nd}, \tag{19a}$$

$$B_i = \Delta_i \otimes b_i \in \mathbb{R}^{nd \times d}, \tag{19b}$$

$$C_i = \mathbb{I}_d \otimes c_i^\top \in \mathbb{R}^{d \times nd}, \tag{19c}$$

with $\Delta_i = \mathrm{diag}(\mathrm{softplus}(W_\Delta(W_u u_i) + b_\Delta)) \in \mathbb{R}^{d \times d}$, $b_i = W_B u_i \in \mathbb{R}^n$, $c_i = W_C u_i \in \mathbb{R}^n$, and $W_u \in \mathbb{R}^{p \times d}$, $W_\Delta \in \mathbb{R}^{d \times p}$ are weight matrices with $p < d$, and $b_\Delta \in \mathbb{R}^d$ is a bias. Note that in formulation (19) the dimensions of the matrices are $\Lambda_i \in \mathbb{R}^{nd \times nd}$, $B_i \in \mathbb{R}^{nd \times d}$, $C_i \in \mathbb{R}^{d \times nd}$, i.e., $n$ is the state dimension and $d$ is the input dimension in the original formulation (4).

### 3.2.3 Recurrent Neural Networks

Given their recurrent nature, one can express LSTMs (5) in the DSF with some basic algebraic manipulations (see Appendix F for details). Once again, we slightly rewrite the standard representation since our goal is to obtain mathematical insights as opposed to computational efficiency.

**qLSTM.** In order to write the qLSTM formulation (8) in the DSF (11), a small modification is needed. In particular, the tanh functions in the input pre-processing (8) and output gate (5b) need to be dropped. Hence, the reformulated qLSTM in the DSF (11) writes as

$$\Lambda_i = \mathrm{diag}(\sigma(W_f u_i)) \in \mathbb{R}^{d \times d}, \tag{20a}$$

$$B_i = \mathrm{diag}(\sigma(W_i u_i)) \odot W_u \in \mathbb{R}^{d \times d}, \quad C_i = \mathrm{diag}(\sigma(W_o u_i)) \in \mathbb{R}^{d \times d}, \tag{20b}$$

where $W_f, W_i, W_o, W_u \in \mathbb{R}^{d \times d}$ are the learnt parameters in (8). It is important to note that here the dimension of the hidden state $h_i$ is equal to the number of input channels $d$, whereas in attention and SSMs the dimension of the hidden state $h_i$ in the DSF (11) is $nd$. For qLSTMs $n = 1$, which will become relevant in further discussions; we refer to the fact that $n > 1$ as *state expansion*.

**RG-LRU.** Given the similarities of RG-LRU [10] and SSMs, it is straightforward to reformulate it into the DSF (11) without the need for modifications besides simple algebraic manipulations. Hence, the RG-LRU can be expressed in the DSF as

$$\Lambda_i = e^{-cr_i \odot \mathrm{softplus}(A)} \in \mathbb{R}^{d \times d}, \quad B_i = \sqrt{1 - \Lambda_i^2} \odot \mathrm{diag}(\sigma(W_B u_i)) \in \mathbb{R}^{d \times d}, \quad C_i = \mathbb{I}_d, \quad (21a)$$

where $r_i = \mathrm{diag}(\sigma(W_R u_i))$, and the function $\sqrt{1 - \Lambda_i^2}$ is applied elementwise to $\Lambda_i$. Similar to the qLSTM and in contrast with the other models, RG-LRU does not have state expansion, i.e. $n = 1$.

## 4 Architecture Comparison: Theoretical and Experimental Results

In this section, we use the DSF to explore some of the long-standing questions between attention, SSMs, and RNNs. We provide theoretical results and/or numerical experiments to substantiate our claims. The experiments presented below are performed on the multi-query associate recall (MQAR) [24] and long range arena (LRA) [3] benchmarks using the code bases [3] provided with the benchmarks. The complete experimental setup and computational resources used are detailed in Appendices J and K, respectively, and a statistical analysis is provided in Appendix L.

### 4.1 Softmax Attention vs. Separable Attention.

Separable attention is used to avoid computation of the query-key matrix $\mathbf{q}\mathbf{k}^\top$. It allows to compute $\mathbf{k}^\top \mathbf{v}$ before multiplying the queries $\mathbf{q}$, which reduces the computational complexity from quadratic to linear in sequence length. While the DSF shows how separable attention, and in particular kernelized attention can be rewritten as a recurrence (11), such a reformulation is only practical for a finite state dimension. However, in the case of $\mathrm{softmax}(\cdot)$, an infinite-dimensional kernel is needed, i.e., in the DSF, softmax attention requires $n = \infty$. This insight can mathematically explain why the good performance observed for softmax attention can only be approximated by separable attention mechanisms, SSMs, or RNNs; but no other architecture is equivalent. The DSF predicts that softmax can be better approximated by growing $n$, which we show in the following theoretical result.

**Lemma 2.** *For two dynamical systems (11) with hidden state dimensions $N$ and $\bar{N}$ with $N \leq \bar{N}$, the dynamical system of state dimension $\bar{N}$ can always recover the dynamical system with state dimension $N$.*

*Proof.* The result follows from the fact that the first $N$ states and the output in (11) can be chosen to be independent of the additional states. The full proof is given in Appendix D. □

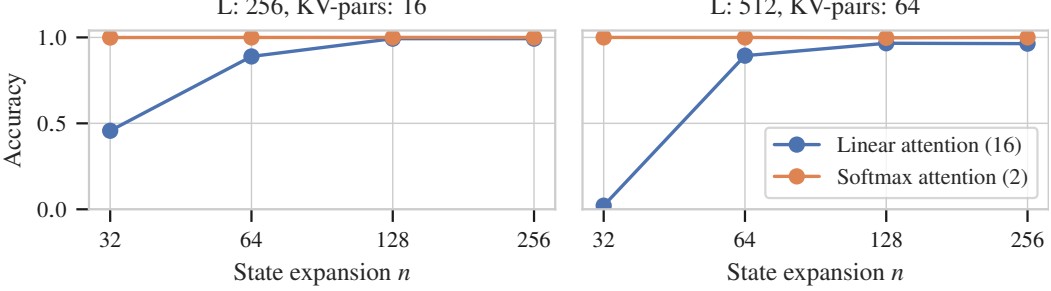

Figure 1: Comparison of linear attention and softmax attention on two MQAR tasks $\{(L = 256, \mathrm{KV\text{-}pairs} = 16), (L = 512, \mathrm{KV\text{-}pairs} = 64)\}$, fixed model size $d = 512$, and varying state expansion $n$. We report the best result from a learning rate sweep in $\mathtt{np.logspace}(-4, -2, 4)$.

Therefore, it holds that the expressivity of a model is non-decreasing with increasing state expansion $n$ (and state dimension $N = nd$), if the rest of the architecture is held constant. As the softmax attention has an infinite hidden state dimension, i.e. $n = \infty$ (Lemma 1), we investigate empirically how its performance compares to linear attention (16), with increasing state dimension on the MQAR. Figure 1 shows that with larger $n$ linear attention converges to the performance of softmax attention, which achieves perfect accuracy throughout.

---

[3]https://github.com/HazyResearch/zoology; https://github.com/google-research/long-range-arena

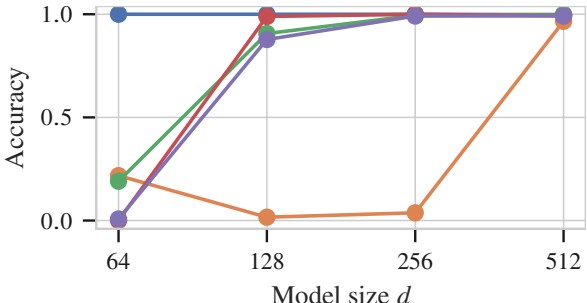

| Model | LRA | WikiText |
|---|---|---|
| Linear Att. (16) | 53.52 | 17.42 |
| Norm. Att. (22) | 58.08 | 16.43 |
| Softmax Att. (2) | 55.96 | 13.15 |
| S6 (4) | 66.84 | N/A |

Figure 2: Model accuracy with increasing model size $d$ for different models: softmax, linear, and normalized attention, S6, and SSD. The MQAR task is $(L = 512, \text{KV-pairs} = 64)$, we fix $n = 128$, and report the best performance of a learning rate sweep in `np.logspace(−4, −2, 4)`.

Table 1: Average accuracy on the LRA benchmark and training perplexity score for different attention architectures (70M params) on the WikiText-103 corpus.

## 4.2 Generalized Linear Attention vs. S6.

By comparing the DSF expressions for both generalized linear attention (15) and S6 (19), we notice that the S6 parameters $b_i = W_B u_i \in \mathbb{R}^n$, $c_i = W_C u_i \in \mathbb{R}^n$ directly correspond to the keys and queries in attention, i.e. $k_i = b_i$ and $q_i = c_i$. Moreover, the state expansion $n$ in S6 is the same as the hidden dimension in attention. However, while this leads to an equivalent output matrix $C_i$ in the DSF parametrization for both architectures, there are remarkable differences between the two:

- **Number of parameters.** The transition matrix $\Lambda_i$ has $d$ parameters in S6 (19) and only 1 in attention. In attention (15), $\eta(q_i, \mathbf{k})$ is the only parameter in $\Lambda_i$, and it is by definition a scalar. In S6, the parameters in $\Lambda_i$ are determined by $\Delta_i \in \mathbb{R}^d$, which has $d$ different parameters. However, it was shown in [8] that the number of parameters in $\Lambda_i$ can be reduced to 1 – similar to attention – without compromising performance. Note that multi-headed attention increases the number of parameters in $\Lambda_i$ from 1 to the number of heads $s$; for more details see Appendix G.

- **Normalization strategy vs Discretization step.** In attention (15), a normalization map $\eta(\cdot)$ is used. This map enters as a fraction in $\Lambda_i$ and also as a denominator in $B_i$. Given that this map is a scalar, these two cancel out when computing the output, as one can see in the convolution representation (12). Hence, in attention $\prod_{k=j}^{i} \Lambda_k B_j$ evaluates to $\frac{1}{\eta(q_i, \mathbf{k}_i)}$. Notice that this does not occur in S6 (19), since the only shared parameter – discretization step $\Delta_i$ – does not cancel out in $\Lambda_i$ and $B_i$ given their different structure. This impacts the selectivity of the matrices on the input, since some input-dependent features are normalized differently in the two architectures.

While the number of parameters in the state transition $\Lambda_i$ does play a role in increasing performance (multi-headed attention typically performs better than single-headed attention [2]), the results in [8] suggest that this role is small. The larger influence thus lies in the recursive structure of $\Lambda_i$ and $B_i$ and/or the parameterization of normalization $\eta(\cdot)$. To further investigate this and the role of normalization in attention, we compare S6 and softmax attention to SSD [8], linear attention [14], and *normalized attention* on the MQAR [24] and LRA [3] benchmarks and train the three attention-based methods on WikiText-103. Inspired by S6, we define *normalized attention* as the attention function

$$\phi(q_i) = q_i, \quad \psi(k_j) = k_j, \quad \eta(u_i) = e^{W_\eta u_i}, \tag{22}$$

where $W_\eta \in \mathbb{R}^{1 \times d}$ is an additional learnt parameter. In Appendix H we discuss two alternatives to (22). The MQAR results are shown in Figure 2 and the average accuracy on the LRA and the training perplexity on WikiText-103 in Table 1. The MQAR results suggest that proper normalization, i.e., using normalization (22), improves the performance of linear attention schemes. This is further supported by the performance of S6 and SSD on the MQAR benchmark, since these two methods also employ input-dependent normalization. Additionally, normalized attention closes part of the gap to softmax attention on the WikiText-103 dataset (Table 1). However on LRA, SSM models (S6) still achieve considerably higher performance than attention-based models. While normalized

attention outperforms linear and softmax attention on the LRA, it performs significantly worse than S6. This result suggests that while the S6 inspired normalization helps to improve performance, the remaining performance gap is possibly explained by the recurrent normalization strategy employed by selective SSM models. Overall these results warrant further research into normalization strategies for attention-based models to explain the performance difference to SSMs. The complete experimental results on the MQAR and LRA benchmarks are detailed in Appendices L and M, respectively.

### 4.3 RNNs vs. S6

Comparing RNNs and S6, it is immediate to observe several similarities. In particular, as shown in Appendix I, the state transition matrix $\Lambda_i$ in S6 (19) can be rewritten (assuming $A = a \cdot \mathbb{I}_{nd}$) as

$$\Lambda_i = \mathrm{diag}(\sigma_{\mathrm{rev}}(\bar{W}_\Delta u_i)^a) \otimes \mathbb{I}_n. \tag{23}$$

Notice that when no state expansion is considered, i.e., $n = 1$ and $\mathbb{I}_n = 1$, this expression almost coincides with the qLSTM state transition (20a), with the only difference that (I) it uses the reversed sigmoid function instead of a sigmoid for the forget gate, and (II) there is an additional learnt parameter $a$ in the exponent. Inspired by the subtle difference in the state transition, we compare the original qLSTM state transition (8) and the S6-inspired state transition 23 on the MQAR benchmark. The performance of both models is shown in Figure 3. We note that the reversed sigmoid state transition outperforms the original state transition on all three benchmark tasks, i.e., the performance of qLSTMs can be improved by insights from S6. Considering state expansion ($n > 1$) for RNNs, the recent xLSTM paper [12] shows that state expanded LSTMs[4] can yield similar performance to S6. This aligns with Lemma 2 and further highlights the importance of state expansion for expressivity. In qLSTM and RG-LRU, state expansion can be easily incorporated by changing the dimensions of the projections $W_f$, $W_i$, $W_o$, where the $\odot$ operation in RG-LRU would be replaced by blockwise operations $\otimes$. Finally, the most apparent difference between the two RNN variants – qLSTM and RG-LRU – and S6 is the parameter coupling in $\Lambda_i$ and $B_i$. While qLSTM does not use a coupling, the couplings in RG-LRU and S6 are performed with different nonlinearities, which is discussed in more detail in [10, Appendix A].

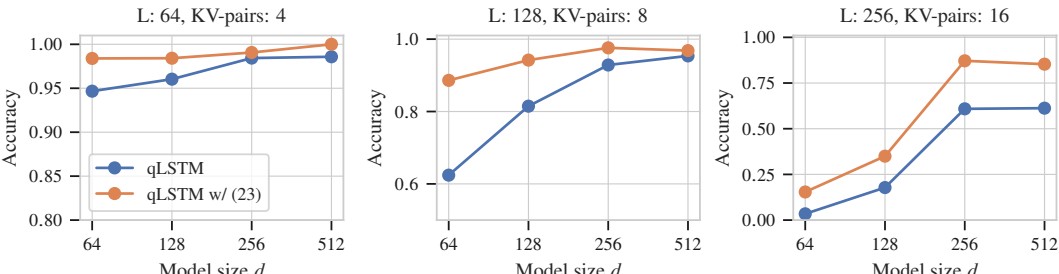

Figure 3: Comparison of qLSTM (8) and a qLSTM variant where the original state transition $\Lambda_i$ is replaced by (23).

## 5 Related Work

State-space models emerged from the S4 architecture by Gu et al. [25], who developed a new theoretically principled approach to sequence modeling rooted in polynomial approximation theory [26]. The result is a transformer-like architecture [2], where attention is replaced by a linear recurrent neural network with special reparametrization. The design of S4 got later simplified in [4, 27], achieving state-of-the-art performance on the long-range arena (LRA) [3] with a highly efficient recurrent mechanism leveraging convolutional views [28], or parallel scans [5, 6].

The high efficiency of SSMs (linear processing) makes them particularly appealing when compared to softmax attention-based transformers, where both inference time and memory suffer quadratically from sequence length. The S4 architecture found first successful applications in reinforcement

---

[4]The presented state expanded LSTM versions, cannot be directly translated into the DSF framework, since the gates not only depend on the inputs $u_i$ but also on past outputs $y_{i-1}$. However, the used state expansion is essentially $n = d$, hence leading to a DSF system of size $d^2$.

learning [29], vision [30], audio [31] as well as online learning [32]. Initial attempts in language modeling [33, 34], supported by theoretical investigations [35, 36] hint at some necessary architectural improvements to unlock the NLP domain. Leveraging in-context learning arguments, a few works [37–39] started incorporating input selectivity mechanisms [40] into SSMs. These efforts culminated in the Mamba architecture [7], which proposed a highly efficient and light (in terms of parameters) input selectivity strategy, with drastic improvements when comparing to earlier variants (H3 [33] and Hyena [41]) on text. This approach was also shown to be effective at byte level [42]. Beyond text, Mamba was recently applied to the vision domain [43, 44] – with outstanding results compared to ViTs [45] both in terms of performance and efficiency. Other applications include e.g. genetics [46], and point clouds [47]. Further, improvements on architectural aspects were proposed in [8, 11, 48].

The design of Mamba is also strongly supported by theoretical evidence showing precisely its superior expressive power compared to S4 [49]. This boost in computational power is due to Mamba's novel input selectivity mechanism resembling gating, which unlocks content-dependent reasoning [7, 40]. Interestingly, input selectivity brings SSMs closer to attention: in particular, Ali et al. [50] showed that the particular parametrization of Mamba can be linked to a non-normalized softmax operator. This finding is also supported by evidence from language theory – Mamba and Attention can solve a similar class of problems [51]. Beyond Ali et al. [50] the connection between linear attention and linear RNNs has been illustrated a few times in the literature [14, 23, 52, 53]. Connections between these architectures have also been carried out using tools from communication complexity in [54, 55]. Compared to these works and to Ali et al. [50], this paper offers a more careful comparison identifying some precise distinctions between SSMs, linear, and softmax attention – which play a nontrivial role in practice and can help bring to light interesting architectural improvements.

## 6 Conclusion

In this paper we presented the DSF, a framework based on dynamical-systems theory that allows analysis of different deep learning architectures by writing them as linear recurrences in state space. We first showed how to reformulate different architectures into the DSF, and then explored (theoretical and experimental) insights resulting from this analysis, thereby answering the questions posed in the introduction. For instance, we showed that with proper normalization the performance of linear attention can be significantly increased (see Fig. 2). We also show, that the DSF allows to integrate insights from one architecture to another as exemplified by Section 4.2. Additionally, the DSF naturally allows analysis of the eigenvalues of the state transition matrix $A$, which are linked to the exploding/vanishing gradient problem [6]. In the case of SSMs and RNNs, the eigenvalues are constrained to be stable by construction, for attention-based models this is not the case and stability needs to be obtained via normalization. The eigenvalues together with the state expansion also affect a model's long-term memory [6]. Both of these aspects can be analyzed via the DSF and should be further investigated in future work. While the training dynamics (especially convergence) can be studied empirically using experiments, the DSF also allows a theoretical analysis. As discussed in Example 2 of [56], a gradient-based optimization algorithm (e.g. SGD) can be interpreted and written as a dynamical system. Using this viewpoint together with the DSF allows interpretation of the training dynamics as two interacting dynamical systems. Therefore, the training dynamics can be theoretically analyzed using tools from control theory, e.g., via Lyapunov theory for convergence and stability of the training. However, we believe this question requires an in-depth investigation and additional empirical validation of the theoretical findings. We expect that the DSF can serve as a tool for principled analysis and design of deep learning architectures.

**Limitations.**  In terms of limitations, it is important to highlight that, while the DSF parametrization allows for a principled comparison between frameworks, architectures written in the DSF do not necessarily enjoy an efficient implementation unless their specific structure can leverage some of the existing algorithms (parallel scan, etc.). In terms of experiments, the insights mentioned above are only verified on two synthetic tasks (MQAR/LRA) and a smaller language task (WikiText-103). To strengthen the insights, a more detailed analysis is needed on larger and more complex tasks.

## Acknowledgments and Disclosure of Funding

Carmen Amo Alonso was partially supported by an ETH AI Center Postdoctoral Fellowship.

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

# A   Visual Representation of the matrix dimensions

Figure 4 represents the dimensions of the recurrence expressed by the linear structured time-varying (LTV) dynamical system described in (11):

$$h_i = \Lambda_i h_{i-1} + B_i u_i,$$

where $h_i \in \mathbb{R}^N$ is the hidden state, $\Lambda_i \in \mathbb{R}^{N \times N}$ is the diagonal state transition matrix, and $B_i \in \mathbb{R}^{N \times d}$ is the input matrix. We highlight the role of the state expansion, where $u \in \mathbb{R}^d$ and $h \in \mathbb{R}^N = \mathbb{R}^{nd}$.

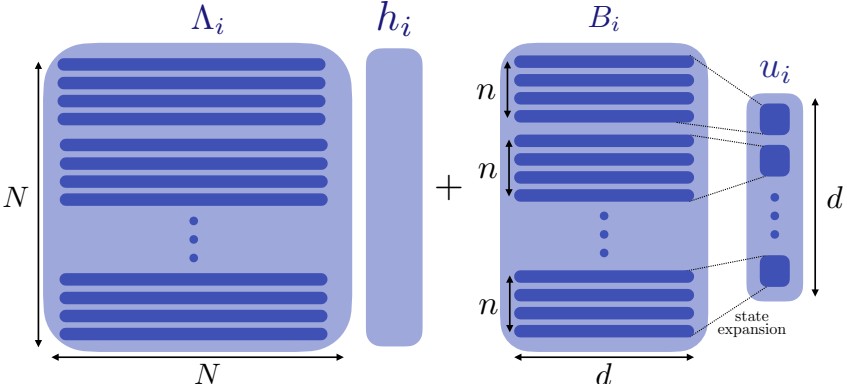

Figure 4: Visual representation of the dimensions considered in the DSF.

# B   Derivation of Separable Attention into DSF

We consider the layer

$$y_i = \sum_{j=0}^{i} f(q_i, k_j, \mathbf{k}_i) W_V u_j,$$

where we define the sequence of keys as

$$\mathbf{k}_i = \{k_0, \ldots, k_i\}.$$

We show that this layer can be equivalently written as the LTV system (11), i.e.,

$$y_i = \sum_{j=0}^{i} C_i \left( \prod_{k=j+1}^{i} \Lambda_k \right) B_j u_j,$$

with $\Lambda_i \in \mathbb{R}^{nd \times nd}$, $B_i \in \mathbb{R}^{nd \times d}$ and $C_i \in \mathbb{R}^{d \times nd}$, if the function $f(\cdot, \cdot)$ is separable as follows

$$f(q_i, \mathbf{k}_i) = \frac{\phi(q_i)^\top \psi(k_j)}{\eta(q_i, \mathbf{k}_i)},$$

where $\phi(q_i) \in \mathbb{R}^n$, $\psi(k_j) \in \mathbb{R}^n$, and $\eta(q_i, \mathbf{k}_i) \in \mathbb{R}$ can be considered to be a normalization function. If the unnormalized part of $f(\cdot, \cdot)$ is a kernel function, it holds that $\psi(k_j) = \phi(k_j)$ for some, possibly infinite dimensional, feature vector $\phi$.

We can compare the output formulations

$$y_0 = C_0 B_0 u_0$$

and

$$\begin{aligned} y_0 &= f(q_0, k_0, \mathbf{k}_0) W_V u_0 \\ &= \phi(q_0)^\top \frac{1}{\eta(q_0, \mathbf{k}_0)} \psi(k_0) W_V u_0, \end{aligned}$$

resulting in

$$B_0 = \left(\frac{1}{\eta(q_0, \mathbf{k}_0)}\mathbb{I}_d \otimes \psi(k_0)\right) W_V \in \mathbb{R}^{nd \times d}, \quad C_0 = \mathbb{I}_d \otimes \phi(q_0)^\top \in \mathbb{R}^{d \times nd}$$

Simlarly, we have

$$y_1 = C_1 B_1 u_1 + C_1 \Lambda_1 B_0 u_0$$
$$y_1 = f(q_1, k_1, \mathbf{k}_1) W_V u_1 + f(q_1, k_0, \mathbf{k}_1) W_V u_0$$
$$= \phi(q_1)^\top \frac{1}{\eta(q_1, \mathbf{k}_1)}\psi(k_1) W_V u_1 + \phi(q_1)^\top \frac{1}{\eta(q_1, \mathbf{k}_1)}\psi(k_0) W_V u_0$$
$$\Rightarrow B_1 = \left(\frac{1}{\eta(q_1, \mathbf{k}_1)}\mathbb{I}_d \otimes \psi(k_1)\right) W_V, \; C_1 = \mathbb{I}_d \otimes \phi(q_1)^\top \text{ and } \Lambda_1 = \frac{\eta(q_0, \mathbf{k}_0)}{\eta(q_1, \mathbf{k}_1)}\mathbb{I}_{nd}$$

and

$$y_2 = C_2 B_2 u_2 + C_2 \Lambda_2 B_1 u_1 + C_2 \Lambda_2 \Lambda_1 B_0 u_0$$
$$y_2 = f(q_2, k_2, \mathbf{k}_2) W_V u_2 + f(q_2, k_1, \mathbf{k}_2) W_V u_1 + f(q_2, k_0, \mathbf{k}_2) W_V u_0$$
$$y_2 = \phi(q_2)^\top \frac{1}{\eta(q_2, \mathbf{k}_2)}\psi(k_2) W_V u_2 + \phi(q_2)^\top \frac{1}{\eta(q_2, \mathbf{k}_2)}\psi(k_1) W_V u_1 + \phi(q_2)^\top \frac{1}{\eta(q_2, \mathbf{k}_2)}\psi(k_0) W_V u_0$$
$$\Rightarrow B_2 = \left(\frac{1}{\eta(q_2, \mathbf{k}_2)}\mathbb{I}_d \otimes \psi(k_2)\right) W_V, \; C_2 = \mathbb{I}_d \otimes \phi(q_2)^\top \text{ and } \Lambda_2 = \frac{\eta(q_1, \mathbf{k}_1)}{\eta(q_2, \mathbf{k}_2)}\mathbb{I}_{nd}.$$

Plugging this back in, we observe

$$y_2 = C_2 B_2 u_2 + C_2 \Lambda_2 B_1 u_1 + C_2 \Lambda_2 \Lambda_1 B_0 u_0$$

$$= \mathbb{I}_d \otimes \phi(q_2)^\top \left(\frac{1}{\eta(q_2, \mathbf{k}_2)}\mathbb{I}_d \otimes \psi(k_2)\right) W_V u_2$$

$$+ \mathbb{I}_d \otimes \phi(q_2)^\top \frac{\eta(q_1, \mathbf{k}_1)}{\eta(q_2, \mathbf{k}_2)}\mathbb{I}_{nd} \left(\frac{1}{\eta(q_1, \mathbf{k}_1)}\mathbb{I}_d \otimes \psi(k_1)\right) W_V u_1$$

$$+ \mathbb{I}_d \otimes \phi(q_2)^\top \frac{\eta(q_1, \mathbf{k}_1)}{\eta(q_2, \mathbf{k}_2)}\mathbb{I}_{nd} \frac{\eta(q_0, \mathbf{k}_0)}{\eta(q_1, \mathbf{k}_1)}\mathbb{I}_{nd} \left(\frac{1}{\eta(q_0, \mathbf{k}_0)}\mathbb{I}_d \otimes \psi(k_0)\right) W_V u_0$$

$$= \phi(q_2)^\top \frac{1}{\eta(q_2, \mathbf{k}_2)}\psi(k_2) W_V u_2 + \phi(q_2)^\top \frac{1}{\eta(q_2, \mathbf{k}_2)}\psi(k_1) W_V u_1 + \phi(q_2)^\top \frac{1}{\eta(q_2, \mathbf{k}_2)}\psi(k_0) W_V u_0$$

$$= f(q_2, k_2, \mathbf{k}_2) W_V u_2 + f(q_2, k_1, \mathbf{k}_2) W_V u_1 + f(q_2, k_0, \mathbf{k}_2) W_V u_0$$

This generalizes the dynamical system matrices to

$$B_i = \left(\frac{1}{\eta(q_i, \mathbf{k}_i)}\mathbb{I}_d \otimes \psi(k_i)\right) W_V,$$
$$C_i = \mathbb{I}_d \otimes \phi(q_i)^\top,$$
$$\Lambda_i = \frac{\eta(q_{i-1}, \mathbf{k}_{i-1})}{\eta(q_i, \mathbf{k}_i)}\mathbb{I}_{nd}.$$

## C   Proof of Lemma 1: Derivation of Softmax Attention into DSF

The softmax function $\mathbb{R}^{n \times m} \to (0, 1]^{n \times m}$ is defined through row-wise normalization and is given by

$$\text{softmax}(\mathbf{z}) := \left(\begin{bmatrix} \frac{e^{z_{0,0}}}{\sum_{j=0}^{m-1} e^{z_{0,j}}} & \cdots & \frac{e^{z_{0,m}}}{\sum_{j=0}^{m-1} e^{z_{0,j}}} \\ \vdots & \ddots & \vdots \\ \frac{e^{z_{n,0}}}{\sum_{j=0}^{m-1} e^{z_{n,j}}} & \cdots & \frac{e^{z_{n,m}}}{\sum_{j=0}^{m-1} e^{z_{n,j}}} \end{bmatrix}\right).$$

The attention block is given as

$$y_i = \sum_{j=0}^{i} \zeta_{i,j}(\mathbf{q}^\top \mathbf{k}) v_j = \sum_{j=0}^{i} \text{softmax}_{i,j}(\mathbf{u} W_Q W_K^\top \mathbf{u}^\top) W_V u_j,$$

where $\text{softmax}_{i,j}(\cdot)$ refers to the element $(i,j)$ of the corresponding matrix. Note that $e^{q_i^\top k_j}$ is a kernel, implying that softmax attention can be brought into the separable form (13). In order to provide the separable form of $e^{q_i^\top k_j}$, we first consider the Taylor expansion of the kernel, which is given by

$$e^{q_i^\top k_j} = \sum_{p=0}^{\infty} \frac{(q_i^\top k_j)^p}{p!}.$$

Each polynomial $(q_i^\top k_j)^p$ represents itself a homogeneous polynomial kernel and its decomposition into a feature vector of $\binom{n+p-1}{p}$ monomials, as shown in [57], is given by

$$\tilde{\phi}_p(x) = \left[ \sqrt{\frac{p!}{n_1! n_2! \cdots n_n!}} x_1^{n_1} \cdots x_n^{n_n} \right]_{n_i \geq 0, \sum_i n_i = p}. \tag{24}$$

The feature representation of the exponential kernel is therefore

$$e^{q_i^\top k_j} = \left[ 1, q_i, \sqrt{\tfrac{1}{2!}} \tilde{\phi}_2(q_i)^\top, \sqrt{\tfrac{1}{3!}} \tilde{\phi}_3(q_i)^\top, \ldots \right] \left[ 1, k_j, \sqrt{\tfrac{1}{2!}} \tilde{\phi}_2(k_j)^\top, \sqrt{\tfrac{1}{3!}} \tilde{\phi}_3(k_j)^\top, \ldots \right]^\top \tag{25}$$
$$:= \phi(q_i)^\top \phi(k_j).$$

Note that to attain the monomials in (24) for a given $p$, one can also use $\bigotimes_{j=1}^p x$, as given in, e.g., [58], which is equivalent up to the constant coefficients, such that we can use $\sqrt{\tfrac{1}{p!}} \tilde{\phi}_p(x) = c_p \bigotimes_{j=1}^p x$, where $c_p$ is a matrix of the respective coefficients multiplying each monomial.

## D    Proof of Lemma 2

Given two dynamical systems of the form (11), we denote the system of hidden state dimension $N$ with the state $h_i^N$ and the matrices $\Lambda_i^N$, $B_i^N$, $C_i^N$ and $D_i^N$ and correspondingly, the system of hidden state dimension $\bar{N}$ using the state $h_i^{\bar{N}}$ and the matrices $\Lambda_i^{\bar{N}}$, $B_i^{\bar{N}}$, $C_i^{\bar{N}}$ and $D_i^{\bar{N}}$. We show that the system of dimension $\bar{N} \geq N$ can recover the system of dimension $N$ by selecting the following system matrices:

$$\Lambda_i^{\bar{N}} = \begin{bmatrix} \Lambda_i^N & 0 \\ \bar{\Lambda} & \hat{\Lambda} \end{bmatrix}, \ B_i^{\bar{N}} = \begin{bmatrix} B_i^N \\ \bar{B} \end{bmatrix},$$
$$C_i^{\bar{N}} = \begin{bmatrix} C_i^N & 0 \end{bmatrix}, \ D_i^{\bar{N}} = D_i^N.$$

It can be seen that the $N$ first states of the system with dimension $\bar{N}$ propagate equivalently to the states of the system of dimension $N$. The additional states evolve independently given any matrices $\bar{\Lambda}, \hat{\Lambda}, \bar{B}$ of appropriate dimension and do not affect the output, such that both systems are equivalent. The two outputs are then equivalent by setting the corresponding entries of the $C_i^{\bar{N}}$ matrix to 0.

## E    Derivation of S6 into DSF

While $A$ in (4) is represented as a dense matrix of size $n \times d$ purely for computational reasons, mathematically $A$ is a diagonal matrix of size $nd \times nd$. This is evident from the fact that S6 parameterizes a different submatrix $A^d \in \mathbb{R}^{n \times n}$ for each embedding dimension $d$, leading to

$$A = \begin{bmatrix} A^1 & & \\ & \ddots & \\ & & A^d \end{bmatrix}.$$

To compute $\Lambda_i$ in (11), the matrix $A$ is multiplied with the selective discretization time $\Delta_i \in \mathbb{R}^{d \times d}$, which is computed as

$$\Delta_i = \text{diag}(\text{softplus}(W_\Delta(W_u u_i) + b_\Delta)), \tag{26}$$

where $W_u \in \mathbb{R}^{p \times d}$, $W_\Delta \in \mathbb{R}^{d \times p}$ are weight matrices with $p < d$, and $b_\Delta \in \mathbb{R}^d$ is a bias. Note that we embed the computed discretization times in a diagonal $d \times d$ matrix to simplify the next reformulations. The product of $\Delta_i$ and $A$ is performed along the embedding dimension axis, i.e.,

$$\begin{bmatrix} \Delta_i^1 A^1 & & \\ & \ddots & \\ & & \Delta_i^d A^d \end{bmatrix} = (\Delta_i \otimes \mathbb{I}_n) \odot A. \tag{27}$$

To arrive at the DSF formulation (19), it only remains to take the exponential function of (27) and state $B_i$, $C_i$ as in (4) with the appropriate dimensions.

## F   Derivation of LSTMs into DSF

### F.1   RG-LRU

In order to replace the abundant elementwise operations $\odot$ in LSTMs with more suitable matrix-vector multiplications for SSMs, we rely on the following observation for $a_i \in \mathbb{R}^d$, $u_i \in \mathbb{R}^d$:

$$\sigma(a_i) \odot u_i = \begin{bmatrix} \sigma(a_i^1) u_i^1 \\ \vdots \\ \sigma(a_i^d) u_i^d \end{bmatrix} = \begin{bmatrix} \sigma(a_i^1) & & \\ & \ddots & \\ & & \sigma(a_i^d) \end{bmatrix} \begin{bmatrix} u_i^1 \\ \vdots \\ u_i^d \end{bmatrix} = \mathrm{diag}(\sigma(a_i)) u_i.$$

As with S6 in Appendix E, we reformulate some quantities for easier presentation, namely we embed the input-dependent vectors $\sigma(W_R u_i) \in \mathbb{R}^d$, $\sigma(W_B u_i) \in \mathbb{R}^d$, where $\sigma(\cdot)$ denotes the sigmoid function, in a diagonal matrix, i.e.,

$$r_i = \mathrm{diag}(\sigma(W_R u_i)) = \begin{bmatrix} r_i^1 & & \\ & \ddots & \\ & & r_i^d \end{bmatrix}, \qquad b_i = \mathrm{diag}(\sigma(W_B u_i)) = \begin{bmatrix} b_i^1 & & \\ & \ddots & \\ & & b_i^d \end{bmatrix}.$$

The DSF representation (21) is then obtained in a straightforward manner.

### F.2   qLSTM

We start by using the same reformulation of the gates as in RG-LRU above:

$$f_i = \mathrm{diag}(\sigma(W_f u_i)), \qquad i_i = \mathrm{diag}(\sigma(W_i u_i)), \qquad o_i = \mathrm{diag}(\sigma(W_o u_i)),$$

where $f_i$ is commonly called the forget gate, $i_i$ and $o_i$ are called the input and output gates, respectively, and $W_f, W_i, W_o \in \mathbb{R}^{d \times d}$. By removing the tanh activation function, we effectively eliminated the input activation gate, which now serves as a standard input to the recurrence (11), i.e., we reformulated the standard qLSTM (8) to the LTV

$$x_i = f_i x_{t-1} + (i_i \odot W_u) u_i$$
$$y_i = o_i h_i.$$

## G   Dynamical System Derivation of Multi-headed Separable Attention

As in Appendix B, we assume a separable attention function, i.e.,

$$f(q_i, k_j, \mathbf{k}_i) = \frac{\phi(q_i)^\top \psi(k_j)}{\eta(q_i, \mathbf{k}_i)}.$$

Additionally, we consider the multi-headed setting introduced in [2, Section 3.2.2], i.e., $s$ different attention operations are performed in parallel. Due to the right multiplication of the input (instead of left multiplication as in the original paper), the output of the different heads is stacked row-wise instead of column-wise. Additionally, we assume there is no output mapping after the attention

operation. This yields the simplified multi-headed layer

$$
y_i = \sum_{j=0}^{i} \begin{bmatrix} [\frac{\phi(q_i)^\top \psi(k_j)}{\eta(q_i,\mathbf{k}_i)} v_j]^1 \\ \vdots \\ [\frac{\phi(q_i)^\top \psi(k_j)}{\eta(q_i,\mathbf{k}_i)} v_j]^s \end{bmatrix} = \sum_{j=0}^{i} \begin{bmatrix} [\frac{\phi(q_i)^\top \psi(k_j)}{\eta(q_i,\mathbf{k}_i)}]^1 & & \\ & \ddots & \\ & & [\frac{\phi(q_i)^\top \psi(k_j)}{\eta(q_i,\mathbf{k}_i)}]^s \end{bmatrix} \begin{bmatrix} [v_j]^1 \\ \vdots \\ [v_j]^s \end{bmatrix}
$$

$$
= \sum_{j=0}^{i} \begin{bmatrix} [\phi(q_i)^\top]^1 & & \\ & \ddots & \\ & & [\phi(q_i)^\top]^s \end{bmatrix} \begin{bmatrix} [\frac{1}{\eta(q_i,\mathbf{k}_i)}]^1 \cdot \mathbb{I}_{n/s} & & \\ & \ddots & \\ & & [\frac{1}{\eta(q_i,\mathbf{k}_i)}]^s \cdot \mathbb{I}_{n/s} \end{bmatrix}
$$

$$
\cdot \begin{bmatrix} [\psi(k_j)]^1 & & \\ & \ddots & \\ & & [\psi(k_j)]^s \end{bmatrix} \begin{bmatrix} [v_j]^1 \\ \vdots \\ [v_j]^s \end{bmatrix},
$$

where $\cdot^s$ denotes the head index. As is standard for multi-headed attention, we reduce the dimensions of the queries, keys, and values by the number of heads, i.e., $q_i \in \mathbb{R}^{m/s}$, $k_j \in \mathbb{R}^{m/s}$, and $v_j \in \mathbb{R}^{d/s}$. Since the $s$ different values $[v_j]^s$ are stacked, this is equivalent to the single headed version, i.e.,

$$
\begin{bmatrix} [v_j]^1 \\ \vdots \\ [v_j]^s \end{bmatrix} = v_j = W_V u_j.
$$

Above observation is also valid for the queries and keys, i.e., we can e.g. write $[\phi(q_i)]^s$ using an indicator function $\mathcal{I}_s(\cdot)$ on the single headed query $q_i$, i.e.,

$$
[\phi(q_i)]^s = \phi(\mathcal{I}_s(q_i)) = \phi(\mathcal{I}_s(W_Q u_i)).
$$

This shows the intuition behind multi-headed attention, which essentially compares parts of the single-headed queries and keys in parallel. Therefore, we can use Appendix B to write multi-headed separable attention in the DSF as

$$
\Lambda_i = \mathrm{diag}\left( \frac{[\eta(q_{i-1},\mathbf{k}_{i-1})]^1}{[\eta(q_i,\mathbf{k}_i)]^1}\mathbb{I}_{d/s}, \ldots, \frac{[\eta(q_{i-1},\mathbf{k}_{i-1})]^s}{[\eta(q_i,\mathbf{k}_i)]^s}\mathbb{I}_{d/s} \right) \otimes \mathbb{I}_n \in \mathbb{R}^{nd\times nd}, \tag{28a}
$$

$$
B_i = \left[ \mathrm{diag}\left( \frac{1}{[\eta(q_{i-1},\mathbf{k}_{i-1})]^1}\mathbb{I}_{d/s}, \ldots, \frac{1}{[\eta(q_{i-1},\mathbf{k}_{i-1})]^s}\mathbb{I}_{d/s} \right) \otimes \psi(\mathcal{I}_s(k_i)) \right] W_V \in \mathbb{R}^{nd\times d}, \tag{28b}
$$

$$
C_i = \mathbb{I}_d \otimes \phi(\mathcal{I}_s(q_i))^\top \in \mathbb{R}^{d\times nd}. \tag{28c}
$$

Multiple heads therefore extend the single scalar in $\Lambda_i$ (in the single-headed case) to $s$ different scalars, however these only act upon a part of the queries $q_i$ and keys $k_j$ due to the indicator function.

## H   Alternative Normalization Schemes

For all experiments in Section 4.2, we use the normalization scheme in (22). The exponential normalization function $\eta(u_i)$ is inspired by softmax attention and S6, which both use exponential functions for normalization (see (18) and (19)). However, other normalization functions can also be considered e.g.

$$
\eta(u_i) = \mathrm{softplus}(W_\eta u_i), \tag{29}
$$

$$
\eta(u_i) = \sigma(W_\eta u_i), \tag{30}
$$

where $\sigma(\cdot)$ denotes the sigmoid function. Table 2 shows an experimental comparison of the exponential normalization function (22) and the two alternatives (29), (30) on the LRA `Image` and MQAR ($L = 512$, KV-pairs $= 64$) tasks. All three normalization schemes perform similarly on both tasks, however the exponential normalization (22) yields the best performance, which is the reason we choose it for normalized attention throughout the paper.

Table 2: Accuracy of the three normalization functions (22), (29), (30) on LRA `Image` and MQAR ($L = 512, \text{KV-pairs} = 64$)

| Normalization Function | Task [%] | |
|---|---|---|
| | LRA `Image` | MQAR ($L = 512, \text{KV-pairs} = 64$) |
| Exponential (22) | 35.96 | 85.9 |
| Softplus (29) | 35.27 | 84.3 |
| Sigmoid (30) | 35.80 | 84.7 |

## I  S6 uses reversed sigmoid in state transition matrix

In the following, we show that the state transition matrix $\Lambda_i$ in S6 is essentially a reversed sigmoid of the projected input. To show this, we assume for simplicity that $A$ in $\Lambda_i = e^{-(\Delta_i \otimes \mathbb{I}_n) \odot A}$ is a scalar, i.e., $A = a \cdot \mathbb{I}_{nd}$. This assumption simplifies $\Lambda_i$ to

$$\Lambda_i = e^{-a(\Delta_i \otimes \mathbb{I}_n)} = \begin{bmatrix} e^{-a\Delta_i^1 \cdot \mathbb{I}_n} & & \\ & \ddots & \\ & & e^{-a\Delta_i^d \cdot \mathbb{I}_n} \end{bmatrix}, \tag{31}$$

where each $e^{-a\Delta_i^j \cdot \mathbb{I}_n}$ itself is a diagonal matrix with $n$-times $e^{-a\Delta_i^j}$ on its diagonal. In order to analyze this expression, we simplify the computation of $\Delta_i$ by fusing the two projection matrices with out loss of generality, i.e.,

$$\Delta_i = \text{diag}(\text{softplus}(W_\Delta(W_u u_i) + b_\Delta))$$
$$= \text{diag}(\text{softplus}(\bar{W}_\Delta u_i)) = \begin{bmatrix} \text{softplus}(\bar{W}_\Delta^{1,:} u_i) & & \\ & \ddots & \\ & & \text{softplus}(\bar{W}_\Delta^{d,:} u_i) \end{bmatrix},$$

where $\bar{W}_\Delta^{j,:}$ denotes the $j^{\text{th}}$ row of $\bar{W}_\Delta$. Above reformulation is valid since the $\text{softplus}(\cdot)$ function is applied elementwise and we note that $\Delta_i^j = \text{softplus}(\bar{W}_\Delta^{j,:} u_i)$ in (31). Using the definition of the $\text{softplus}(\cdot)$ function, we can show that

$$e^{-a\Delta_i^1} = e^{-a\,\text{softplus}(\bar{W}_\Delta^{j,:} u_i)} = (1 + e^{\bar{W}_\Delta^{j,:} u_i})^{-a} = \sigma_{\text{rev}}(\bar{W}_\Delta^{j,:} u_i)^a, \tag{32}$$

where $\sigma_{\text{rev}}(\cdot)$ is the reversed sigmoid, i.e., $\sigma_{\text{rev}}(x) = \frac{1}{1+e^x}$. Since the reversed sigmoid is again applied elementwise to a vector or a matrix, we can write the S6 state transition matrix as

$$\Lambda_i = \text{diag}(\sigma_{\text{rev}}(\bar{W}_\Delta u_i)^a) \otimes \mathbb{I}_n,$$

where the power $a$ is applied elementwise. The assumption $A = a \cdot \mathbb{I}_{nd}$ we made in the beginning, can be relaxed to any diagonal matrix, however the resulting $\Lambda_i$ will have a more complex representation.

## J  Experimental Details

The experimental results provided in Section 4 are performed on the multi-query associative recall (MQAR) benchmark [24], the long range arena (LRA) benchmark [3], and the WikiText-103 dataset. To obtain the MQAR and LRA results, we modified the Zoology[5] and LRA[6] code bases and added the normalized attention model and the selective SSM models, respectively. The code for both benchmarks is provided on GitHub for MQAR[7] and for LRA[8] separately.

---

[5] https://github.com/HazyResearch/zoology
[6] https://github.com/google-research/long-range-arena
[7] https://github.com/IntelligentControlSystems/dsf-mqar
[8] https://github.com/jsie7/ssm-benchmark

## J.1 MQAR experiments

**Training Details**   We evaluate the following three architecture classes:

1. **Attention:** softmax attention [2], linear attention [14], normalized attention (22). For all three attention functions, we use a standard GPT-2 style multi-headed Transformer architecture, where we replace the attention block with the respective attention function. The three attention functions are defined in Section 2.1. For all MQAR runs we use a single attention head.

2. **State space model:** S6 [7], SSD [8]. For both SSM variants, we use a standard GPT-2 style single-headed Transformer architecture, where we replace the attention block with the respective SSM variant. This means for S6 and SSD, we do not implement the pre-convolution on the input or the SiLU activations; but just the S6 and SSD blocks. We do this to ensure a fair comparison of the backbones (sequence mixers) irrespective of the higher-level architecture. The S6 and SSD blocks are defined in Section 2.2 and we use the provided code base[9] to implement it.

3. **RNN:** qLSTM [9], modified qLSTM. We embed both qLSTM variants in a standard GPT-2 style single-headed Transformer architecture, where we replace the attention block with the qLSTM. We do this to ensure a fair comparison of the backbones (sequence mixers) irrespective of the higher-level architecture. The standard qLSTM is defined in Section 2.3 and the modified qLSTM is the same as the standard qLSTM but with modified state transition (23), i.e., a modified forget gate.

For all MQAR runs, we use the following training protocol:

- **Optimizer and schedule:**  Weight decay of 0.1, linear warmup with duration of 10%, AdamW optimizer [59].   For each run, we sweep the learning rates in `np.logspace`$(-4, -2, 4)$ and train for 64 epochs. This is the same setup as in [24].

- **Training duration:** We use a global batch size of $512$, which we reduce to $256$ if sequence length $L \geq 128$, to $128$ if sequence length $L \geq 256$, and to $64$ if sequence length $L \geq 512$. We do this to keep the memory consumption approximately constant over different tasks.

- **Width and depth:** For all runs, we use two layers (each with a sequence model and a MLP, interleaved with layer normalization). The model dimensions $d$, state expansion $n$, sequence length $L$, and number of KV pairs are swept according to the experiment (see Section 4). This is the same setup as in [24].

- **Position information:** Positional embeddings [60] are used for the attention and RNN architecture classes, but not for the SSM architecture classes. This is the same setup as in [24].

- **Data:** Each model is trained on 100,000 datapoints and evaluated on 3,000 datapoints. The data and its order are constant for all runs. This is the same setup as in [24].

**Performed Experiments**   We run the three attention models and the two state space models on four different MQAR tasks, i.e., $\{L = 64, \text{KV-pairs} = 4\}$, $\{L = 128, \text{KV-pairs} = 8\}$, $\{L = 256, \text{KV-pairs} = 16\}$, and $\{L = 512, \text{KV-pairs} = 64\}$, which progressively increase in complexity. For each model and task, we sweep both the model size $d = [64, 128, 256, 512]$ and the state expansion $n = [32, 64, 128, 256]$,[10] resulting in a total of 320 experiments. We only report the results of the best performing learning rate; the full results of all experiments are stated in Appendix L.
We run the two qLSTM variants on three different MQAR tasks, i.e., $\{L = 64, \text{KV-pairs} = 4\}$, $\{L = 128, \text{KV-pairs} = 8\}$, and $\{L = 256, \text{KV-pairs} = 16\}$. For both variants we sweep the model size $d = [64, 128, 256, 512]$, resulting in a total of 24 experiments. We only report the results of the best performing learning rate; the full results are reported in Figure 3.

---

[9]`https://github.com/state-spaces/mamba`

[10]We did not increase $n$ further, since the selective scan CUDA kernel provided for S6 and SSD is capped at $n = 256$; for more information see `https://github.com/state-spaces/mamba`.

## J.2 LRA experiments

**Training Details**   We evaluate the following two architecture classes:

1. **Attention:** softmax attention [2], linear attention [14], normalized attention (22). For all three attention functions, we use a standard GPT-2 style multi-headed Transformer architecture, where we replace the attention block with the respective attention function. The three attention functions are defined in Section 2.1. To ensure a fair comparison, we keep all hyperparameters of the three attention models constant except the attention function.

2. **State space model:** S6 [7]. We use a standard GPT-2 style single-headed Transformer architecture, where we replace the attention block with the S6 block. This means, we do not implement the pre-convolution on the input or the SiLU activations; but just the S6 block. We do this to ensure a fair comparison of the backbones (sequence mixers) irrespective of the higher-level architecture. The S6 block is defined in Section 2.2 and we use the provided code base[11] to implement it.

For all LRA runs, we use the following training protocol:

- **Optimizer and schedule:** Linear warmup with duration of 10% and AdamW optimizer [59].
- **Position information:** Positional embeddings [60] are used for the attention architecture classes, but not for the SSM architecture classes.
- **Data:** Each model is trained on the standard datasets provided with the LRA benchmark.

The exact hyperparameters for each LRA task and each model are reported in the publicly available code base.[12] Note that we do not optimize the hyperparameters, i.e., the reported accuracies might be lower than in the original LRA paper [3].

**Performed Experiments**   We run the three attention models and the S6 models on the LRA tasks `ListOps`, `Text`, `Retrieval`, `Image`, and `Pathfinder-32`, which are summarized below; for the full details we refer to [3].

1. List Operations (`ListOps`): This task evaluates a model's ability to capture hierarchical dependencies over long contexts. The goal is to predict the result of a mathematical operation consisting of nested *mean*, *median*, *max*, and *min* operations,[13] The task is a ten-way classification task with maximal input lengths of 2k.

2. Text Classification (`Text`): This task evaluates a model's ability to capture the tone of long tokenized texts. The dataset consists of IMDb movie reviews, which need to be classified as negative or positive in tone. The task is a binary classification task with maximal input lengths of 4k.

3. Document Retrieval (`Retrieval`): This task evaluates a model's ability to compress long sequences into representations that are suitable for similarity matching. The dataset consists of tokenized papers published by the American Academy of Neurology (AAN), which need to be classified in having a citation link or not. The task is a binary classification task with maximal input lengths of 8k.

4. Image Classification (`Image`): This task evaluates a model's ability to learn 2D spatial relations from a 1D vector. The dataset consists of vectorized images, which depict one of ten possible classes, e.g. a horse or a car. The task is a ten-way classification task with maximal input lengths of 1k.

5. Long-Range Spacial Dependency (`Pathfinder-32`): This task evaluates a model's ability to learn spacial dependencies in a vectorized image. The dataset consists of images, which depict two circles and multiple dashed paths. The goal is to evaluate whether the two circles are connected by any of the present paths or not. The task is therefore a binary classification task with maximal input lengths of 2k.

---

[11]https://github.com/state-spaces/mamba

[12]https://github.com/jsie7/ssm-benchmark

[13]For instance, `input:  max(4, min(5,6, mean(9, 4, 5)))`, `output:  5`.

## J.3 WikiText-103 experiments

**Training Details** We use the 70M parameter Pythia architecture (Pythia70M) [61].[14] For softmax attention we use the standard Pythia attention block, while for linear attention [14] and normalized attention (22) we replace the attention block in the Pythia architecture with the respective attention functions defined in Sections 2.1 and 4.2.

For all training runs on WikiText-103, we use the following protocol:

- **Optimizer and schedule:** Weight decay of 0.1, linear warmup with duration of 10%, AdamW optimizer [59] with $\beta = (0.9, 0.95)$, and gradient clipping $= 1$. For each run, we sweep the learning rates in $[0.0003, 0.001, 0.003, 0.01]$ and train for 50 epochs.
- **Training duration:** We use a batch size of 128 and train for 50 epochs.
- **Width and depth:** We use a context length of 1024 and the standard Pythia70M configuration, i.e., model size of 512, 8 heads, and 6 layers.
- **Position information:** Positional embeddings [60] are used as in standard Pythia.

**Performed Experiments** We train the three attention functions on WikiText-103 and sweep the learning rates $[0.0003, 0.001, 0.003, 0.01]$. For all three attention functions learning rate 0.003 performed best and the corresponding results are reported in Table 1.

# K Computational Resources

All experiments (MQAR, LRA, and WikiText-103) were run on a cluster with 11 nodes with the following GPU and CPU specifications:

| GPU Model | Nr. of nodes | memory/GPU | GPUs/node | CPUs/node |
| --- | --- | --- | --- | --- |
| NVIDIA GTX 1080 Ti | 1 | 11 GB | 8 | 20 |
| NVIDIA GTX 2080 Ti | 2 | 11 GB | 8 | 64 |
| NVIDIA GTX 3090 | 1 | 24 GB | 8 | 128 |
| NVIDIA GTX 4090 | 1 | 24 GB | 8 | 128 |
| NVIDIA TITAN RTX | 1 | 24 GB | 8 | 128 |
| NVIDIA Quadro RTX 6000 | 1 | 24 GB | 8 | 128 |
| NVIDIA V100 | 2 | 32 GB | 8 | 44 |
| NVIDIA A100 | 1 | 40 GB | 8 | 48 |
| NVIDIA A100 | 1 | 80 GB | 10 | 48 |

The MQAR and LRA training and test runs were parallelized and assigned to the best available GPU node, while the parallelized training on WikiText-103 was exclusively run on the NVIDIA A100 (80GB) node.

For each learning rate sweep of the MQAR runs described in Appendix J we estimate the average runtime to be 1h,[15] leading to a total unparallelized runtime of 54 days for all MQAR tasks. There where approximately 20 runs for debugging and training purposes, which were terminated after a few minutes, thus we did not include them in the time estimate.

For each task of the LRA benchmark, we estimate the average runtime to be 2h for `Image`, 5h for `Text`, 6h for `ListOps`, 30h for `Retrieval`, and 45h for `Pathfinder`, leading to a total unparallelized runtime of 31 days for all LRA tasks. Note that to obtain better hyperparameters, they would need to be tuned for each task, which would significantly increase the total runtime. There where approximately 30 runs for debugging and training purposes, which were terminated after a few minutes, thus we did not include them in the time estimate.

---

[14]https://github.com/EleutherAI/pythia

[15]Obviously, larger model sizes and MQAR tasks with larger sequence length took longer than smaller models and tasks with shorter sequence length. However, a more accurate time estimate is hard to obtain due to the cluster setup with multiple different GPU models and the fact that we terminated tasks early if the 99% accuracy threshold was achieved.

For the training on WikiText-103, each learning rate sweep took approximately 14h, leading to a total parallelized runtime of 42h for all three attention models. We estimate a total of 1h of runtime for tuning runs, bringing the total runtime to 43h.

## L    Extended Results on MQAR

Figures 5 and 6 show the MQAR results of Figures 2 and 3 but for multiple runs over 10 different seeds.

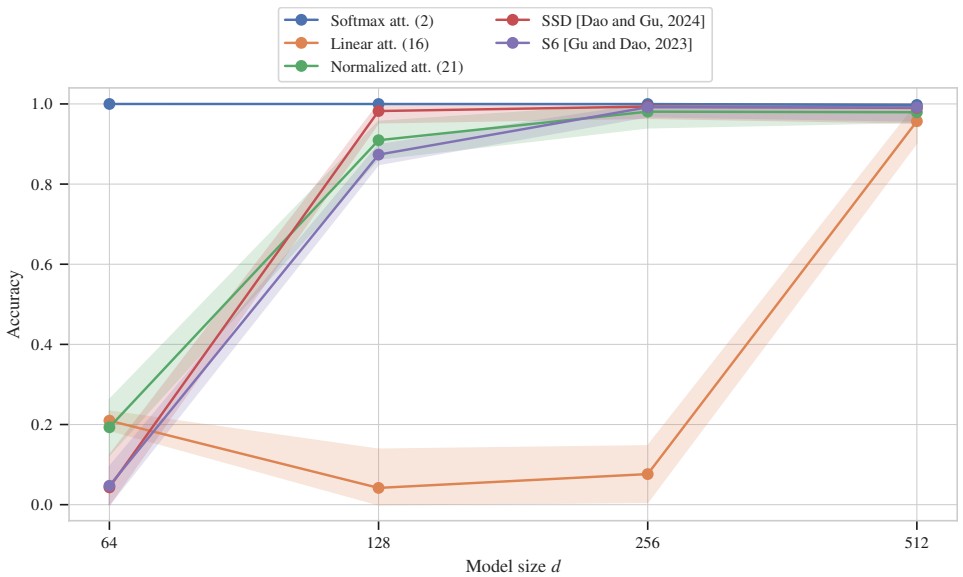

Figure 5: Model accuracy with increasing model size $d$ for different models: softmax, linear, and normalized attention, S6, and SSD. The MQAR task is ($L = 512$, KV-pairs $= 64$), we fix $n = 128$, and report the best performance of a learning rate sweep in `np.logspace`$(-4, -2, 4)$. Solid lines are the average accuracy over 10 different seeds, while the shaded area show the standard deviation.

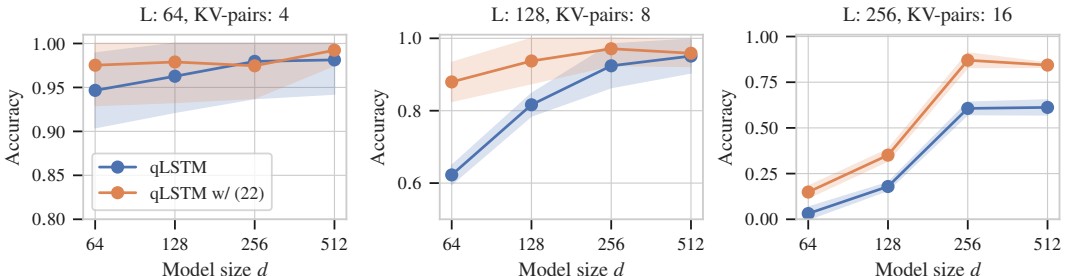

Figure 6: Comparison of qLSTM (8) and a qLSTM variant where the original state transition $\Lambda_i$ is replace by (23). Solid lines are the average accuracy over 10 different seeds, while the shaded area show the standard deviation.

In Figure 7 we report the complete results of all MQAR experiments detailed in Appendix J. A selected subset of these are already presented in Figure 1 and Figure 2 in the main text.

The effect of state expansion can not only be observed for linear attention (Figure 1) but also for normalized attention (22), S6, and SSD for the task $\{L = 512, \text{KV-pairs} = 64\}$. Contrary to this, for small model sizes $d$ and larger tasks (e.g. normalized attention, task $\{L = 512, \text{KV-pairs} = 64\}$, $d = 64$) the performance decreases with increased state expansion $n$ or shows erratic behaviour. Since this behavior only occurs for small $d$, we hypothesis that this effect is due to the model being to small to accurately learn the task.

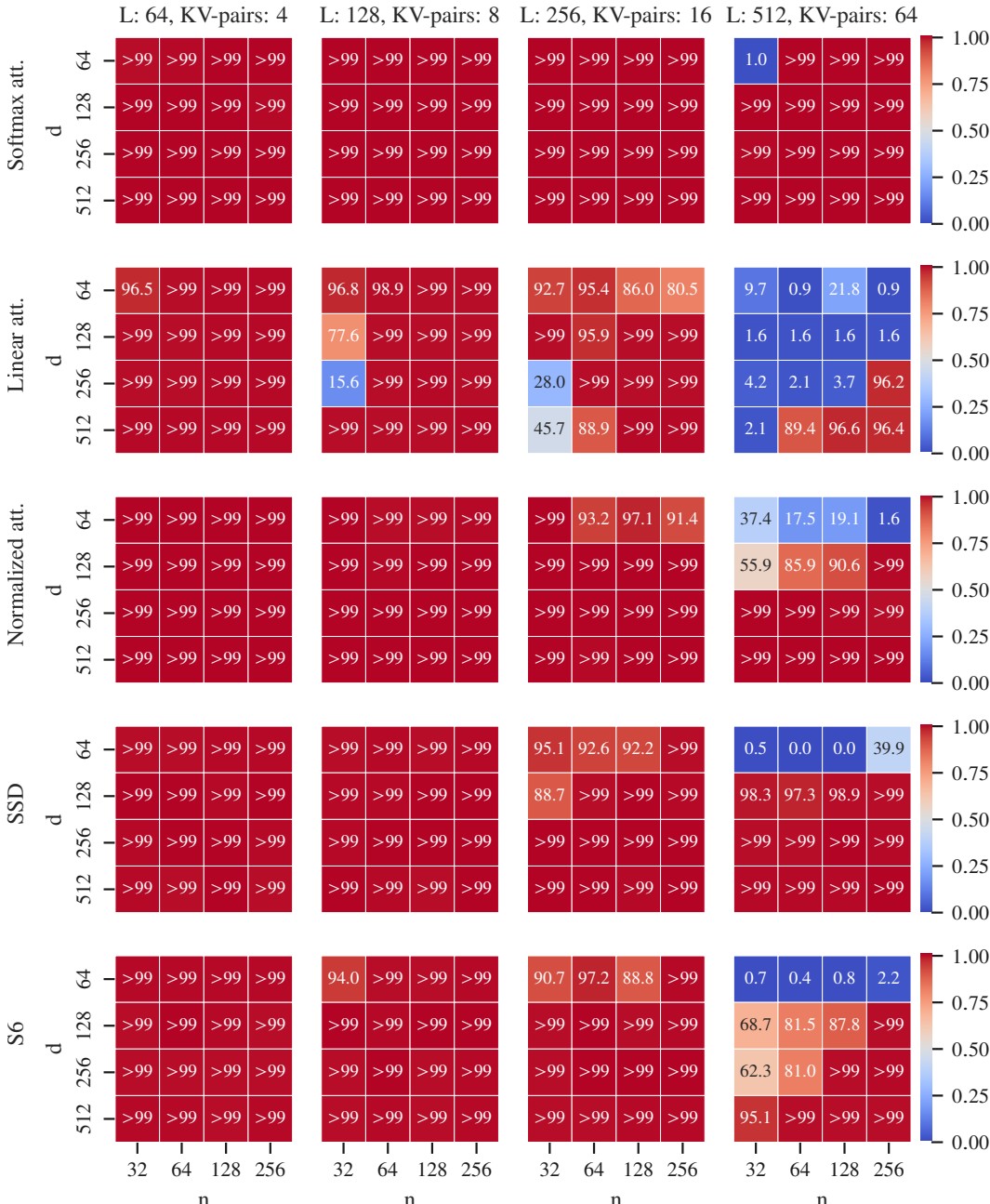

Figure 7: Results for softmax attention [2], linear attention [14], normalized attention (22), S6 [7], and SSD [8] on four different, progressively harder MQAR tasks $\{L = 64,$ KV-pairs $= 4\}$, $\{L = 128,$ KV-pairs $= 8\}$, $\{L = 256,$ KV-pairs $= 16\}$, and $\{L = 512,$ KV-pairs $= 64\}$. We sweep the model size $d = [64, 128, 256, 512]$ and the state expansion $n = [32, 64, 128, 256]$ for each model and task. We only report the best performance from a learning rate sweep in `np.logspace(−4, −2, 4)` measured as accuracy on the MQAR task. The accuracy is denoted in % in the grid in the figure.

While Figure 2 shows that normalized attention outperforms standard linear attention [14], Figure 7 shows an even more significant performance gap. Additionally, we note that SSD outperforms S6, which was already hinted at in [8], and that normalized attention performs on par with S6. Together these results hint at the importance of normalization both for attention and SSMs. The comparison of S6 and SSD shows that reducing the number of parameters in the state transition $\Lambda_i$ from $d$ to a scalar does not hurt performance, which is further supported by the findings in [8]. These experiments

also suggest that the recursive structure in $\Lambda_i$ and $B_i$ (present in S6 and SSD but not in normalized attention or linear attention) is less important than proper normalization of the attention scores. Additionally, the results on WikiText-103 (Table 1) show that better normalization can close 25% of the perplexity gap between linear attention and softmax attention. Together, these results warrant more investigations into new and better normalization techniques for attention-based models.

Finally, we remark that softmax attention performs perfectly accross all sweeps except $\{L = 512, \text{KV-pairs} = 64\}$, $d = 64$, and $n = 32$, which is most likely due to a too small model or too small learning rate.

## M Extended Results on LRA

In Table 3 we report the complete results of all LRA experiments detailed in Appendix J. The average performance over all tasks is reported in Table 1 in the main text.

While normalized attention (22) performs slightly better on LRA than the other two attention-based models, there is a significant performance gap to the S6 model (a selective SSM variant). The reason for this gap potentially lies in the recurrent normalization employed in selective SSMs (see Section 4.2). Interestingly, S6 only achieves significantly higher accuracy on the tasks Text and Image, showing that selective SSM models are particularly suited for long-range classification of text and image modalities.

Table 3: Model performance in terms of test accuracy on the LRA benchmark. The first entry (Random) represents the performance of random guessing on the task, i.e., indicating the baseline above which a model is considered to have learned a meaningful representation.

| Model | LRA Task [%] | | | | | |
|---|---|---|---|---|---|---|
| | ListOps | Text | Retrieval | Image | Pathfinder | avg |
| Random | 10.00 | 50.00 | 50.00 | 10.00 | 50.00 | 34.00 |
| Softmax Attention [3] | 35.72 | 63.10 | 77.46 | 34.22 | 69.32 | 55.96 |
| Linear Attention [14] | 16.12 | 64.41 | 76.44 | 37.97 | 72.64 | 53.52 |
| Normalized Attention (22) | 38.24 | 64.96 | 79.68 | 35.96 | 71.56 | 58.08 |
| S6 [7] | 38.02 | 81.34 | 80.50 | 65.08 | 69.26 | 66.84 |

