# OpenReview forum: "Understanding the Differences in Foundation Models: Attention, State  Space Models, and Recurrent Neural Networks"
_NeurIPS.cc/2024/Conference — NeurIPS 2024 poster_

### Official Review · Reviewer_BUkP · 2024-06-23

**Soundness:** 3
**Presentation:** 3
**Contribution:** 2
**Rating:** 6
**Confidence:** 4

**Summary:**

The paper concerns using the proposed "Dynamical Systems Framework" (DSF) to better understand the differences between linear SSMs, RNNs, linear attention and Softmax attention. The DSF shows a way to write each of these models as a linear time-varying recurrence. The paper aims to use this to answer questions about the differences between the three types of models, the role of state expansion, and how ideas from SSMs can be used to improve RNNs. Empirical experiments on the multi-query associative recall (MQAR) task and Wikitext are used to support the findings.

**Strengths:**

- The goal of better understanding the differences between the different models, and in particular trying to understand the gap between Softmax attention and its more efficient alternatives is important
- The paper is well-written and clear. I particularly liked Figure 4 in Appendix A which made the dimensions being referred to throughout the paper clear.
- The different formulations and results in the paper appear to be technically correct
- Insights are used from SSMs to propose a different normalization scheme for linear attention which appears to improve performance on the tasks considered
- The empirical results on the MQAR and Wikitext experiments support the claims

**Weaknesses:**

- The proposed DSF is nice in that it allows comparing each of the different types of models, but it's originality is weakened by the fact that most of the main parts of the framework have been discussed before in prior work
  - The connections between linear attention and linear RNNs/SSMs are clear and have been discussed in prior works as mentioned in related work.
  - In addition, the infinite dimensional representation of softmax attention as discussed in Section 3.2.1 is also discussed in Katharopoulos et al. 2020 among others
  - Softmax attention has also been formulated as a dynamical system in https://arxiv.org/abs/2401.06104 which should be cited
- The findings that are suggested as a result of the framework also do not seem to be new or particularly surprising
  - The fact that larger state sizes lead to increased expressivity is not surprising and has been explored before, e.g. in https://arxiv.org/abs/2312.04927, https://arxiv.org/abs/2312.00752
  - The proposed improvement to RNNs is to change the parameterization of a linear RNN (the quasi LSTM) by replacing its transition parameterization with that of S6. But this seems to just make it more like the already existing linear RNN, RG-LRU (with a slight different in parameterization from S6). This seems to be a much weaker finding than the claimed "What do selective SSMs teach us about improving RNN architectures" from line 53 of the Intro.
   - As a counter to this point I am making, the proposed improvement to the normalization of linear attention appears to be novel and appears to improve performance on the tasks considered.
- The focus and claims around expressivity and performance in the paper appear to be very language focused.
  - This is obviously of interest, but perhaps narrows the ability for the DSF to provide new and interesting insights as I mention in the point above.
  -  In addition, only 2 small tasks are considered to support the claims.  This limits the ability to understand if the claims hold in more general settings.
  - The authors state in the limitations that to strengthen the insights, a larger and more complex language task is needed. This could be great, however I would suggest perhaps also considering more diverse data modalities could also strengthen the insights of the proposed framework.

**Questions:**

- Perhaps considering data modalities and examples other than language could help to highlight insights that can be drawn from the DSF? E.g. perhaps modalities that more naturally arise from dynamical systems could help to further highlight differences between softmax attention and its more efficient alternatives? Or could help highlight differences between different linear attention/SSM/RNN variants?

- Are the differences between softmax attention and the other methods on the MQAR task, e.g. in Figure 5, simply due to the difference in state size? One can think of the state size of softmax attention as the size of its KV cache required for the sequence length of the task. If you make the recurrent alternatives have a state size that matches the softmax attention KV cache size for the task sequence length, do they then perform as well? Or are there other things happening in Softmax Attention as well? Perhaps exploring this through the lens of the DSF framework could also provide insights?

Minor:
- It is stated the Matrix in Equation 12 is $L\times L$, but isn't this only the case when $B$ and $C$ are $N \times 1$ and $1 \times N$? The presentation in the paragraph before suggests these are general matrices.
- Line 203 says "The DSF makes apparent how a necessary condition for separability is
for the map ζ(·) to be expressed by a finite-dimensional kernel, ". But in line 154 it says "Softmax attention also satisfies the assumption of separability".  I realize the point that is trying to be made regarding "practical separability", but nonetheless, it appears to be a contradiction as currently written.

**Limitations:**

Adequately discussed.

---

> ### Author Rebuttal · Authors · 2024-08-07
>
> Thank you very much for the helpful feedback and pointers! Changes or clarifications (as deemed appropriate) for every single point raised will be incorporated in the final version of the paper. In what follows we provide a brief discussion on the points that were raised in the review:
>
> **Prior studies:** Thank you for the feedback and pointing us at [Oren et al. (2024)](https://arxiv.org/abs/2401.06104). We agree that connections between the three model classes have been studied in previous work and are well-established. Our intent is to translate the three model classes specifically as dynamical systems - which is achieved by the DSF - in order to use methods from control theory to analyze the models and to obtain a single form of representation to easily compare certain parameterization choices between the model classes. In the revised version, we made sure to better highlight this intent and we appropriately cited [Oren et al. (2024)](https://arxiv.org/abs/2401.06104).
> We also acknowledge that the infinite representation of softmax-attention has been discussed previously in [Katharopoulos et al. (2020)](https://arxiv.org/abs/2006.16236) but also in [Nauen et al. (2024)](https://arxiv.org/abs/2403.02920) and [Choromanski et al. (2021)](https://arxiv.org/abs/2009.14794) among others. We believe the discussion on representation of softmax-attention is important in the context of the DSF, which is why we included it. In the revised manuscript, we made sure to include additional related works and improve the presentation to better reflect previous works.
>
> **Empirical novelty:** We acknowledge that the insight on increased performance with larger state sizes is well-established in the literature, we believe it is beneficial to show that insights provided by the DSF are consistent with previous results. In the revised manuscript, we improved the presentation of this insight to clearly state that we are only recovering existing results and cite the relevant works more prominently.
> For the insights on RNNs, the main difference between the proposed qLSTM variant and the RG-LRU is the shared parameterization in state transition $A$ and input matrix $B$. While RG-LRU shares parameters in both matrices, the qLSTM does not (as briefly mentioned in lines 263-265). Interestingly, any attention-based model also shows this coupling in the DSF. In the revised manuscript, we added a discussion of this point for all three model classes (please also see the global response) and plan to include an empirical analysis for the camera-ready version.
>
> **Task complexity & modalities:** Thank you for the feedback and the suggestion. We agree that more complex and also diverse tasks can strengthen the insights we provide. While we believe in-depth analysis of a specific insight (e.g. the proposed normalized attention) including showing performance on complex tasks are more suited for subsequent works (we only have limited computational resources), we agree that other task modalities offer more insights. Therefore, to improve the empirical validation of our findings, we include experiments on the LRA benchmark, which also contains image tasks, in the revised manuscript. As an example, we included the results of the LRA image task in the attached pdf. In doing this we generated an additional insight (performance gap between SSM models and transformers on LRA) as discussed in the general response.
>
> **Attention and state size:** Thank you for the excellent suggestion. Preliminary experiments we ran show better performance of the other models with their state size increased to the size of the KV cache, but not outperforming softmax attention. Hence, it seems that there is more to softmax attention than just the size of the KV cache. Additionally, we believe there is more potential to such an analysis, since it relates to the question of how information is compressed in the state of the DSF. This is important since running these models with the size of the KV cache is not sustainable in practical applications with larger sequence lengths. We think such an analysis could lead to further follow-up works investigating different aspects of this question. Therefore, we will conduct full experiments for the revised version and perform an in-depth analysis of the results.
>
> **Minor:** Thank you for pointing out the two inconsistencies. It is true that the matrix in Eq. (12) is not $L \times L$, but should be a block matrix consisting of $L \times L$ blocks. We will correct this in the revised manuscript. We will also improve the language for practical separability, since it is indeed misleading in the current form.

---

> ### Comment · Reviewer_BUkP · 2024-08-12
>
> Thank you for the clarifying comments and additional results. I believe after the response to the reviewers this is a stronger paper and of interest to the community. I have increased my score.

---

> > ### Author Response · Authors · 2024-08-14
> > **Thank you for the review**
> >
> > Thank you for taking the time to review the rebuttal and the positive score. We will make sure to incorporating all of your valuable feedback into the final version of the paper. Your thoughtful input is greatly appreciated and helped us improve the quality of our work.

---

### Official Review · Reviewer_GS8T · 2024-07-08

**Soundness:** 3
**Presentation:** 2
**Contribution:** 2
**Rating:** 6
**Confidence:** 4

**Summary:**

This paper shows that many sequence models (including attention, SSMs, and recurrent models) can be viewed as linear time-varying dynamical systems. This is helpful for answering some questions about the differences and similarities between these architectures.

**Strengths:**

**Importance of unifying framework**. The present state of sequence model research is quite muddy, with tons of new architectures being proposed. On the surface these architectures often look quite complicated and differentiated, when in reality they are closely related to one another.  Because of this, I think unifying frameworks are useful for helping push progress in the space.

**The role of normalization in S6 and Linear Attention.** The paper includes a very interesting discussion of the parameterization of the normalization in S6 and Linear attention.

**Weaknesses:**

**Presentation and clarity.** The introduction and abstract do not provide any details on how the Dynamical Systems Framework works and assumes a reader is familiar with dynamical systems. To improve the presentation and accessibility, I would recommend including a high-level discussion of what the DSF is and what challenges it addresses.

**Theoretical Novelty.** In the first paragraph of the introduction the authors state, “Although these models show great promise in boosting efficiency, current comparisons with attention are merely empirical.” This isn’t correct: there have been many theoretical studies comparing these new recurrent architectures with Softmax attention. To name a few:

- [Arora *et al.](https://arxiv.org/abs/2402.18668)* include a theoretical analysis grounded in results from communication complexity that highlights differences in models ability to perform associative recall.
- Using tools from circuit complexity, [Merill *et al.*](https://arxiv.org/pdf/2404.08819) show that Attention and SSMs are in the same complexity class ($\text{TC}^0$).
- Using tools from communication complexity, [Bhattamishra *et al.*](https://arxiv.org/pdf/2406.09347)  provide separation results between attention and recurrent architectures on numerous tasks.

**Empirical Novelty.** The main empirical result in this paper is that increased state size (across architectures) leads to improved performance on MQAR and that separable attention matches Softmax attention with sufficiently large recurrent size. This result has already been shown theoretically and empirically in [Arora *et al.*](https://arxiv.org/pdf/2402.18668) (ICML ‘24) - see Figure 2, Theorem 3.1, and Section 3. The work under review builds upon Arora *et al.* by including a few additional architectures to the analysis. But this prior work should be mentioned in the related work and/or the introduction given the significant overlap in results .

**Claims around Lemma 2.** Some of the language around Lemma 2 is a bit incautious. The authors state that because of Lemma 2, “the larger the state expansion n, the more expressivity the architecture has”. In other words, they are claiming that increasing state expansion strictly increases expressivity. To support this claim, the theorem would need to additionally show that a dynamical system of state dimension $N$ ***cannot*** recover all dynamical systems with state dimension $\hat{N}$.  Additionally, I think its important to add the qualification that this holds assuming everything else about the architecture is held constant. It’s of course possible for an architecture with smaller state size to admit solutions that a larger state size model cannot represent if the parameterizations are different.

**Minor typos and clarifications.**

- Line 32: “framework that allows to evaluate” → “allows us to evaluate”
- Line 61: “We use $\sigma(\cdot)$ to denote is the sigmoid function.
- In the list of questions answered by the paper (Line 40), I would include hyperlinks to the relevant sections.

**Questions:**

It would help to add additional exposition on the results in Fig. 2. What are the takeaways? It looks like exponentiating improves performance, but this isn’t stated explicitly.

**Limitations:**

Yes the authors discuss limitations.

---

> ### Author Rebuttal · Authors · 2024-08-07
>
> Thank you very much for the helpful feedback and pointers! Changes or clarifications (as deemed appropriate) for every single point raised will be incorporated in the final version of the paper. In what follows we provide a brief discussion on the points that were raised in your review:
>
> **High-level discussion:** Thank you for this suggestion! We indeed believe that this would help to make the paper accessible to a broader audience. To address this comment, we plan to add an introduction to the concept of dynamical systems and how it applies in the context of foundation models. We will also delve into the specific dynamical system parametrization chosen by the DSF, which is the canonical state space representation. This choice will be motivated by two reasons: (i) this exists a wide body of literature on state space model representations for dynamical systems, and (ii) it encompasses transformers, RNNs and SSMs in a suitable fashion that allows for further analysis. We will also better explain how expressing foundation models in the DSF allows for a direct comparison between models and enables the use of control theoretical tools for their study.
>
> **Prior theoretical comparative studies:** Thank you for pointing this out, we acknowledge that the statement in the introduction is not correct. We intended to convey that to the best of our knowledge so far there has been no unified approach to theoretically analyze transformers, SSMs, and RNNs, but only empirical studies benchmarking specific models of these classes against each other. In the revised version, we changed this to reflect prior theoretical work comparing various models with attention, including properly referencing the papers that are mentioned in the review.
>
> **Empirical novelty:** Thank you for pointing us at [Arora et al. (2024)](https://arxiv.org/abs/2402.18668). We agree that the insight “increased state size leads to better performance” is not novel and has been shown in previous work. We intended to include this insight, since we believe it is beneficial to show that insights generated from the DSF are consistent with the literature. In the revised manuscript, we appropriately cite [Arora et al. (2024)](https://arxiv.org/abs/2402.18668) and improve the presentation to reflect that this insight has been reported before. Additionally, we improved the presentation of the overall document to better highlight the other two insights generated from the DSF (i.e. normalized attention and the S6-inspired state transition for qLSTMs) as well as the additional insights we added (please see the global response for more details). This change puts more focus on the new insights and explicitly presents the state size finding as established knowledge.
>
> **Lemma 2:** Thank you for the feedback, we acknowledge that the language in and around Lemma 2 is imprecise. As correctly pointed out, Lemma 2 should state that expressivity with a larger state size is at least as good as with lower state size, i.e., not strictly better. We reformulated Lemma 2 and the text surrounding it to reflect this. Additionally, we now explicitly state that this only holds if everything else is held constant as mentioned in the review.
>
> **Results discussion:** Thank you for the feedback. We agree that the key take-aways from Fig. 2 are lacking and are not appropriately discussed in the text. In the revised manuscript, we added the following discussion points:
> - The results in Fig. 2 and Tab. 1 support the conclusions that (i) performance of linear attention can be improved by using a suitable normalization function and (ii) this normalization function does not need to use the keys and queries but can be learnt independently.
> - The difference in parameterization between the proposed normalized attention and SSD mainly lies in the recurrent nature of the normalization parameters in transition matrix $A$ and input matrix $B$ for normalized attention. With the addition of the LRA experiments in the revised manuscript and the performance gap between SSMs and transformers, this warrants the question on the role of this recurrent normalization. In the revised manuscript we added a discussion on this question as detailed in the global response.
>
> **Typos:** Thank you! We will amend these in the revised version.

---

> > ### Comment · Reviewer_GS8T · 2024-08-13
> >
> > Thank you to the authors for their detailed response. I look forward to reading about the new insights provided by their framework.
> >
> > I raise my score from a 5 to a 6.

---

> > > ### Author Response · Authors · 2024-08-14
> > > **Thank you for the review**
> > >
> > > Thank you for taking the time to review the rebuttal and the positive score. We will make sure to incorporating all of your valuable feedback into the final version of the paper. Your thoughtful input is greatly appreciated and helped us improve the quality of our work.

---

### Official Review · Reviewer_US2F · 2024-07-10

**Soundness:** 2
**Presentation:** 3
**Contribution:** 3
**Rating:** 5
**Confidence:** 3

**Summary:**

This paper provides a dynamical system based framework for principled comparisons between various existing recurrent architectures (linear attention, SSMs, etc.). There are focuses on the formulation and the role of the hidden state dimension. Some experimental results on the multi-query associative recall (MQAR) benchmark and the WikiText-103 dataset are provided to demonstrate the differences between the considered models.

**Strengths:**

- The paper is overall well written and organized
- Systematic comparisons between the formulations of various sequence models are provided, which are neat and useful for researchers without much background in the area

**Weaknesses:**

- The two proposed experiments (the MQAR benchmark and the WikiText-103 dataset) seem random and limited. There are no experiments on the more standard benchmark tasks such as the LRA task (focusing on smaller tasks such as the sequential CIFAR task would be valuable) and time series forecasting tasks
- The theoretical insights are actually quite limited. There are unaddressed natural questions such as:

     (1) one might wonder about finite-dimensional approximation of the softmax attention,

     (2) how Lemma 1 could be related to the kernel method and RKHS

- There are also other aspects that are left out for the comparative studies: e.g., how do the training dynamics differ between the various models (which architecture will converge faster), as well as the stability of the models (how prone are they to the infamous vanishing/exploding gradient problem). Also, can we quantify and compare the long range dependency learning capability of these models?
- No practical guidance in terms of which architecture is suitable for what kind of task is provided. After all, this should be the main goal of the comparisons

**Questions:**

See above

**Limitations:**

Yes

---

> ### Author Rebuttal · Authors · 2024-08-07
>
> Thank you very much for the helpful feedback and pointers! Changes or clarifications for every single point raised will be incorporated in the final version of the paper. We provide a brief discussion on the points that were raised in the review:
>
> **Benchmark:** Our main goal was to show-case some selected insights the DSF can bring to all three investigated model classes, i.e., transformers, SSMs, and RNNs. Since softmax-attention-based models are known to be the best performing models for language modalities and considerable research focuses on sub-quadratic models to close the performance gap to softmax-attention, we opted to benchmark on language tasks. However, we added the LRA benchmark to the experimental validation of our insights (see global response). The LRA results are consistent with existing empirical results, which show that attention-based models perform worse than SSM-based models on the LRA. Why this is the case is still an open research question and warrants more detailed investigation, but the DSF points to a potential hypothesis for this performance gap: the recurrent normalization discussed in Section 4.2 of the paper. Along with the LRA results we added a discussion on this point to the revised paper.
>
> **Insights:** Thank you for your feedback. We agree that the paper only showed a few theoretical insights. Therefore, we have extended on these insights and added several more as discussed in the general response. Also thank you for the specific suggestions, which we discuss below.
> - *Finite-dimensional approximation:* In the revised manuscript, we expanded on the discussion in Section 3.2.1 on using a Taylor approximation of softmax-attention. Specifically for the Taylor approximation, there exist error bounds that can be used to bound the error between the approximation and softmax-attention on a given interval, e.g., the Lagrange bound (see e.g. Section 3.3.2. in [Chevillard et al. (2008)](https://hal-lara.archives-ouvertes.fr/hal-02102827)). The Lagrange error bound defines the worst-case error of the approximation on the given interval and thus provides a quantitative metric of how well softmax-attention can be approximated. Additionally, we extended the discussion of existing works that use a finite-dimensional approximation of softmax.
> - *Kernel method:* Thank you for the excellent question. The attention function $\zeta$ in Eq. (13) (later rewritten in Lemma 1) can be interpreted as a kernel if $\psi = \phi$ , with softmax-attention using a specific type of exponential kernel function, i.e., $\exp(x^\top y)$. This was noticed and further analyzed in [Tsai et al. (2019)](https://arxiv.org/abs/1908.11775) and Katharopoulos et al. (2020). In the revised manuscript, we added a remark on this and refer to the mentioned papers for more details.
>
> **Training dynamics:** While training dynamics (especially convergence) can be studied empirically using experiments, the DSF also allows theoretical analysis. As discussed in Example 2 of [Dörfler et al. (2024)](https://arxiv.org/abs/2401.14029), a gradient-based optimization algorithm (e.g. SGD) can be interpreted and written as a dynamical system. Using this perspective together with the DSF allows interpretation of the training dynamics as two interacting dynamical systems. Thus, the training dynamics can be analyzed theoretically with tools from control theory, e.g., via Lyapunov theory for convergence/stability of the training. However, we believe this question requires an in-depth investigation and additional empirical validation, which are out of scope for this paper.
>
> **Stability:** The exploding/vanishing gradient problem is linked to the eigenvalues of state transition matrix $A$ for SSMs (Orvieto et al. (2023)). Therefore, SSM models actively restrict the eigenvalues to the range [0,1]. Using the DSF the same argumentation can be made for transformers and RNNs. In the revised paper, we included an additional discussion on this.
>
> **Long-range dependencies:** A model's abiltiy to capture long-range dependencies are due to two factors: (i) the modulus of the eigenvalues in dynamic matrix $A$, and (ii) the dimension of the state $x$ (in DSF representation). Other factors include the compression of the input into the state by the encoding architecture, but this is out of the scope of our analysis that focuses on the recurrent block of learning architectures. Given two trained models with numerical values for $A$ and a sufficiently large state dimension, the DSF would be capable of predicting the performance in long-range contexts by looking at the eigenvalues of $A$. Moreover, theory on system theoretic model reduction (e.g. [Obinata & Anderson (2012)](https://dl.acm.org/doi/abs/10.5555/559486)) provides principled ways of comparing the ability of a given state dimension to compress past information. Thus, the DSF can be leveraged to provide a theoretical analysis on the long-range capabilities of the analyzed models. To complement our analysis, we will incorporate this discussion into the revised manuscript.
>
> **Task suitability:** We believe that the question of architecture suitability for specific tasks is a broad open research question, and many factors need to be considered to provide a clear answer. Specifically, the recurrent models presented in this paper are only one aspect of deep learning architectures, but the encoders, decoders, skip connections, etc. also play a role in task suitability. For this reason, we do not aim to answer this question in full in this paper. However, motivated by the comment we have experimentally explored different data modalities. Our findings illustrate (see attached pdf and paper experiments) that transformer architectures (specially with softmax attention) tend to perform better on language tasks and SSMs on tasks that require long-range dependencies such as the LRA benchmark. However, the performance of a model is highly dependent on the specific formulation of the task.

---

> > ### Comment · Reviewer_US2F · 2024-08-13
> > **Thank you for the response**
> >
> > I am satisfied with the response and have raised my score.

---

> > > ### Author Response · Authors · 2024-08-14
> > > **Thank you for the review**
> > >
> > > Thank you for taking the time to review the rebuttal and the positive score. We will make sure to incorporating all of your valuable feedback into the final version of the paper. Your thoughtful input is greatly appreciated and helped us improve the quality of our work.

---

### Official Review · Reviewer_BcKL · 2024-07-13

**Soundness:** 3
**Presentation:** 3
**Contribution:** 3
**Rating:** 7
**Confidence:** 3

**Summary:**

This paper introduces the Dynamical Systems Framework (DSF), a theoretical approach for analyzing and comparing various foundation models in AI. The DSF reformulates attention-based models, State Space Models (SSMs), and Recurrent Neural Networks (RNNs) into a common dynamical systems representation. This allows for principled comparisons and generates new insights into their similarities and differences. The authors leverage the DSF to compare linear attention and selective SSMs, provide conditions for approximating softmax attention, analyze state expansion in RNNs and SSMs, investigate differences between linear attention and S6 (Mamba) models, and apply SSM insights to RNN architectures. The paper combines theoretical results with empirical validations on the MQAR benchmark and WikiText-103 dataset.

**Strengths:**

- Novel unified perspective on different foundation model architectures
- Enables new theoretical insights and comparisons between previously isolated model classes
- Sound theoretical analysis with supporting lemmas and proofs
- Empirical validation on MQAR and WikiText-103 provides practical grounding
- Clear potential for guiding future development of efficient, scalable models
- Innovative application of SSM insights to improve RNN architectures

**Weaknesses:**

- Limited empirical evaluation, focusing primarily on MQAR and a single language modeling task
- Lack of statistical significance reporting for experimental results
- Some theoretical results (e.g., Lemma 2) are relatively straightforward
- Dense theoretical sections may be challenging for broader audience
- Immediate practical impact on model performance is somewhat limited
- Doesn't fully explore implications for training or inference efficiency

**Questions:**

1. How well might the DSF insights scale to much larger models (billions of parameters) used in state-of-the-art applications?
2. Are there theoretical limits to the convergence of linear attention to softmax attention performance with increased state expansion?
3. Have you explored alternative normalization schemes that might bridge the gap between S6 and linear attention while maintaining computational efficiency?
4. How does the S6-inspired state transition in qLSTM affect its long-term memory capabilities?
5. Can the theoretical insights from DSF reformulations be leveraged to develop new, efficient implementations?

**Limitations:**

The authors acknowledge that the DSF parametrization, while theoretically insightful, doesn't necessarily lead to efficient implementations. Empirical validation is limited to synthetic tasks and a relatively small language modeling task, constraining generalizability. The paper also lacks comprehensive analysis of computational efficiency implications for different architectures under the DSF lens.

---

> ### Author Rebuttal · Authors · 2024-08-07
>
> Thank you very much for the helpful feedback and pointers! Changes or clarifications (as deemed appropriate) for every single point raised will be incorporated in the final version of the paper. In what follows we provide a brief discussion on the points that were raised in the review:
>
> **Empirical evaluation:** We agree that the empirical evaluations of our insights were somewhat limited. To improve this we added additional experiments on the LRA benchmark, which also includes image tasks to broaden the empirical evaluation. The partial results (due to time constraints), i.e., only the sequential CIFAR-10 (image) task are shown in Tab. 1 of the attached pdf, and the results of the full benchmark will be added to the revised manuscript.
>
> **Statistical significance:** Thank you for this comment! We are including error bars to Fig. 2 and 3 in the paper. These results are shown (as shaded regions) in Fig. 1 and 2 of the attached pdf.
>
> **Theoretical results:** We acknowledge that Lemma 2 is somewhat straightforward and that this effect has already been shown empirically. However, this finding is a trivial consequence of the DSF and we wanted to highlight that the DSF provides insights that are consistent with the literature.
>
> **Theoretical sections:** Thank you for this comment! We acknowledge that the theoretical sections might be dense for an audience unfamiliar with dynamical systems. To improve upon this point, we plan to incorporate a brief introduction to the conceptual basics of dynamical systems in the introduction of the revised manuscript. We will also extend Section 3.1 with a more detailed presentation to make it more amenable for a broader audience.
>
> **Practical impact:** We added several more insights and analyses to the revised document as stated in the global response and we believe that the proposed work enables leveraging system theoretic insights in order to generate more specific practical impacts in future works.
>
> **Training and inference efficiency:** Although the primary goal of the DSF is to carry out theoretical comparison and obtain insights on the mathematical representations of foundation models, we believe it can also help in providing pointers for efficient training and inference. For instance, as discussed below in the response to the question regarding efficient implementations, the DSF allows identification of efficient algorithms to implement a specific model. Additionally, the DSF naturally allows analysis of the eigenvalues of the state transition matrix $A$, which are linked to the exploding/vanishing gradient problem ([Orvieto et al. (2023)](https://arxiv.org/abs/2303.06349)). We included a discussion on this to the revised version.
> On a more theoretical level, it is possible to view the training process as an interaction of two dynamical systems by formulating the optimization algorithm as a separate dynamical system (see e.g. Example 2 in [Dörfler et al. (2024)](https://arxiv.org/abs/2401.14029)). In the revised version, we added a remark on this, but we think elaborating on this requires a more in-depth analysis that is out of scope of the paper.
>
> **Scalability:** In order to explore how the insights (e.g. normalized attention) scale to billions of parameters, an empirical evaluation is vital. Due to the extensive computational demands of this task, we believe this analysis is out of the scope for this paper. In a theoretical analysis, normalized attention scales linearly with context length, similarly to linearized attention and SSD. As such, we anticipate that it will scale similarly to other SSM models that have been demonstrated at a larger scale.
>
> **Softmax approximation:** Thank you for the excellent question! We believe this is a very interesting research question that warrants further investigation, but has no straightforward answer. However, in the special case of polynomial kernel functions (in Lemma 1), softmax-attention can be exactly recovered using infinitely many polynomials. Therefore, in the limit these polynomial kernels can recover the softmax performance with appropriate weights as there is an exact correspondence.
>
> **Alternative normalization schemes:** Yes, we also considered other normalization schemes to the one shown in Eq. (21), i.e.,
> \begin{align}
> 	\eta(u_i) &= e^{\textrm{ReLU}(W_\eta u_i)} \\\\
> 	\eta(u_i) &= \sigma(W_\eta u_i)
> \end{align}
> We observed that they perform similarly to the one presented in the paper. We included the results on these additional normalization schemes to the appendix of the revised manuscript including a brief discussion.
>
> **Long-term memory:** The long-term memory capabilities of a model from a control theoretic perspective depend on the eigenvalues of the state transition matrix $A$ and the size of the state $x$ and thus the size of $A$ ([Orvieto et al. (2023)](https://arxiv.org/abs/2303.06349)). Therefore, any model reformulated in the DSF can be analyzed in this way. Since the proposed state transition is the same state transition as in S6, the two models have the same theoretical long-term memory capabilities. These capabilities then solely depend on the hyper-parameter $n$ (state size) and the initialization of $A$. However, we agree that this question warrants a more detailed analysis and thorough empirical validation, which is out of scope of this paper.
>
> **Efficient implementations:** Thank you for the excellent question. Yes, the DSF can be leveraged to identify and also develop new efficient implementations. In the revised version, we expand the first paragraph in Section 3.2 that only briefly mentions that the DSF can be used for this. In the appendix, we also added a few examples of how this can be done, e.g., the proposed normalized attention can be efficiently implemented using flash linear attention.

---

> > ### Comment · Reviewer_BcKL · 2024-08-11
> >
> > I thank the authors' for their detailed responses to my and other reviewers' comments. I am satisfied with the answers, and look forward to seeing the final version of the paper. While I appreciate the paper for its theoretical contribution, I personally find theoretical frameworks to be most compelling when used to engineer and design novel architectures and training protocols. In the case of SSMs, I am more willing to make an exception because I believe that the literature in this field is scattered and oftentimes talking past each other. If I can suggest one thing, I would like the authors to really hone the introduction to allow this paper to become a good introductory theoretical framework for understanding these various architectures. I will maintain my positive score.

---

> > > ### Author Response · Authors · 2024-08-14
> > > **Response to Reviewer BcKL**
> > >
> > > Thank you for taking the time to review the rebuttal and the positive score. We will make sure to improve the introduction according to your feedback. Your thoughtful input is greatly appreciated and helped us improve the quality of our work.

---

### Author Rebuttal · Authors · 2024-08-07

We would like to thank all the reviewers for their time and effort evaluating our paper. We believe the insightful reviews helped us to greatly improve the paper. The main contents of the rebuttal are several added insights we gained from the DSF, extended experiments including the LRA benchmark, and several clarifications in the text regarding existing works and theoretical results.

**Additional experiments:** In response to the reviews, we extended the evaluation of our insights to the Long Range Arena (LRA) benchmark. The LRA benchmark adds an additional data modality (images) to the evaluation and also allows us to investigate the well-known performance gap between attention-based models and SSMs. Additionally, we plan to test our insights on time series forecasting, specifically on the Informer ETT benchmark.

**Additional insights gained from DSF:** In response to all the reviews, we added several additional insights generated from the DSF.
The goal of this paper is to introduce the DSF as a useful framework to approach theoretical research questions about foundation models. The results included are meant to exemplify important questions that the DSF can answer and open future research directions, rather than be a comprehensive list of the insights, each of which will require theoretical derivations in their own right. However, in order to address the concern of limited insights and enhance the theoretical contributions of this work, we added the following insights:
- Investigating the performance of attention and selective SSM models on the LRA benchmark, has led us to investigate the performance gap in more detail (SSMs outperform transformers significantly). We found that the gap can be explained by the recurrent normalization strategy (discretization step) used by selective SSMs as discussed in Section 4.2 of the paper.
- Using the DSF, we extended our study to provide an analysis of the eigenvalues of the state transition matrix $A$ of all three model classes. These eigenvalues are directly linked to the exploding/vanishing gradient problem ([Orvieto et al. (2023)](https://arxiv.org/abs/2303.06349)). In the case of SSMs and RNNs, the absolute value of the eigenvalues are constrained in the range $[0,1]$ by construction. We observe that for attention-based models this is also true empirically due to the normalization used.
- In the DSF formulation of softmax-attention it is revealed that attention-based models share parameters in the state transition matrix $A$ and the input matrix $B$. While we mention this fact in the paper (lines 263-265), we expanded the discussion of this since SSMs and RG-LRU also shared this parameterization. However, this is not the case in more standard RNNs.

**Novelty:** In response to the reviews, we clarified our own theoretical contributions and improved the presentation to appropriately reflect existing results. The main contribution of our paper is the introduction of the DSF, which is a unifying framework for analysis of transformers, SSMs, and RNNs. To the best of our knowledge, this is the first unified framework that allows analysis of all three model classes in the same parameterization and thus allows to identify differences in the models that lead to significant performance differences. While some of the provided results already exist in the literature (e.g that increased state size improves performance), we also provide novel insights unique to the DSF framework in a comprehensive way that enables further analysis with control theoretical tools. For instance, the DSF enabled proposing the normalized attention, which we have extended in the revised version.

**Attached pdf:** The attached pdf contains Figures 2 & 3 of the original paper including error bars, i.e., confidence margins over 10 random seeds, which we aim to increase further as computational constraints permit. Additionally, Table 1 in the attached pdf shows preliminary results on the LRA benchmark, specifically for the image task (sequential CIFAR-10).

---

### Decision · Program_Chairs · 2024-09-25

**Decision:**

Accept (poster)

**Comment:**

The authors introduce the Dynamical Systems Framework (DSF), a unified theoretical approach for comparing attention-based models, State Space Models (SSMs), and Recurrent Neural Networks (RNNs) as linear time-varying dynamical systems. This framework offers insights into the similarities and differences among these architectures, particularly in hidden state dimensions and state expansion, and demonstrates its effectiveness with empirical validations on the MQAR benchmark and WikiText-103 dataset. While the paper is strong in its contributions, it could benefit from improved presentation and clarity, and further emphasis on its theoretical novelty, as some may find the analysis results predictable. Nonetheless, the paper’s contributions are significant, and I recommend its acceptance.